# The ANN Architecture Analysis: A Case Study on Daylight, Visual, and Outdoor Thermal Metrics of Residential Buildings in China

**Shanshan Wang [1,2], Yun Kyu Yi [3,*] and Nianxiong Liu [4]**

1   Medical Architecture and Environment Research Unit, School of Architecture and Urban Planning, Beijing University of Civil Engineering and Architecture, Beijing 100044, China; wangshanshan@bucea.edu.cn
2   Beijing Key Laboratory of Green Building and Energy-Efficiency Technology, Beijing 100044, China
3   School of Architecture, University of Illinois Urbana-Champaign, Champaign, IL 61820-5711, USA
4   School of Architecture, Tsinghua University, Beijing 100084, China; phlnx@tsinghua.edu.cn
*   Correspondence: ykyi@illinois.edu

**Abstract:** Selecting an appropriate ANN model is crucial for speeding up the process of building performance simulation during the design phase of residential building layouts, particularly when evaluating three or more green performance metrics simultaneously. In this study, daylight, visual, and outdoor thermal metrics were selected as main green performance. To find the suitable ANN model, sensitivity analysis was used to obtain a set of proper parameters applied to the ANN structure. To train the ANN model with a higher predicting accuracy, this paper tested four different scenarios of ANN parameter setups to find some general guidelines about how to set up an ANN model to predict DF, sunlight hours, QuVue and UTCI. The results showed that an ANN model with a combined output variable demonstrated better average prediction accuracy than ANN models with a separated output variable. Having two times the number of training samplings compared to the number of input variables can lead to a high accuracy of prediction. The ideal number of neurons in the hidden layer was approximately 1.5 times the number of input variables. These findings of how to improve the ANN model may provide guidance for modeling an ANN for building performance.

**Keywords:** artificial neural network (ANN); sensitivity analysis; daylight factor (DF); sunlight hours; QuVue (sky view ratio); universal thermal climate index (UTCI)

## 1. Introduction

Comparing the ANN methods with other types of simulation tools in building performance, the ANN method has some strong points in terms of simplicity, calculation speed, and learning from limited data sets [1–4]. Recently, many papers have used the artificial neural network (ANN) as a prediction model in the field of building performance, for instance, in energy use, daylight, and energy demand [5–12].

Some studies have investigated several kinds of expert-knowledge-based algorithms for ANNs [13,14], including multiple linear regression (MLR), Gaussian process regression (GPR), support vector machine (SVM), boosted tree, random forest, and so on, to compare their performance in the prediction of building performance [15,16]. Each method has its own purpose and feature to accomplish a prediction assignment. Moreover, in each algorithm, there exist some principles to make a better model to achieve ideal prediction outcomes.

The research of the ANN structure has been focused on how to enhance the performance of its models' prediction. Some studies conducted different training algorithms to achieve high accuracy of the ANN model's performance. Kwok et al. revealed that some input variables could significantly improve the accuracy of the model [17]. Mustafaraj et al. found that in building and validating models for predicting dry bulb temperature and

relative humidity for different time steps, when the step-ahead time scales increased, the ANN models had a lower accuracy performance [18]. Jovanović et al. investigated three ANN models (FFNN, RBFN, and ANFIS) to improve heating energy consumption prediction accuracy, and found that the ensemble, by combining the outputs of member networks, achieved better prediction results [19]. Buratti et al. compared the prediction performance of IAQ in naturally ventilated buildings, drawing the conclusion that feed-forward backpropagation neural network models provided a greater accuracy of $R^2$ and a higher-quality prediction with a lower RMSE compared with the MLR models [20]. Deng and Chen conducted five training algorithms according to MAE and $R^2$ and found that Levenberg–Marquardt predicted the thermal comfort in indoor environments in ten offices and ten apartments/houses in Indiana, USA [21]. In past decades, optimal ANN models have been researched deeply.

In the meantime, previous studies related to ANNs have also focused on the principle behind how the input and output variables or the number of neurons or layers were set up, raising questions about constructing optimal ANN models in the area of predicting building performance. The hidden neuron is one of key elements to conducting an optimal ANN model. Yokoyama et al. conducted a study on the relationship between the number of neurons in the hidden layer and the MAE between the predicted and actual thermal sensation to determine a reasonable number of hidden neurons [22]. Moreover, the number of neurons in the input layer and in the hidden layer, the number of sample cases, and the order of inputs were also mostly investigated. Huang et al. employed a forward selection method to choose individual candidate variables. They found that an oversized network with a large input order and number of hidden layers can enhance the prediction error of ANN models with high frequency noise [23]. Rocío Escandón et al. concluded that the sample size ensured reliable prediction results according to a specific building category, finding that a ratio of 2.5 between the sample size and the number of characteristic parameters (input variables) could increase the accuracy [24]. The combination of each element of an ANN model related to building performance has been discussed. Due to the complexity and diversity of ANN prediction problems, it is required to conduct more research on improving the accuracy related to the building area. The determination of the ANN architecture for building energy consumption in order to make a quick prediction was prevalent in many studies [25–28], but in the green building design stage, how to establish the proper architecture of an ANN prediction model for green performance was the focus of only few studies.

In previous studies, the normalized method was commonly conducted to construct the ANN models; however, the normalization combination of the input and output variables was seldom investigated regarding the accuracy performance of the models.

In summary, the ANN is a common machine-learning algorithm that gives a highly accurate prediction performance and is efficient as well as computationally time saving. An in-depth discussion about improving the precision capacity of the ANN model in the studies mentioned above on the ANN structure is also an important area because it is critical for building a highly accurate prediction model. However, most of these papers focused on one factor of the ANN structure, such as the number of input variables or a comparison of one pair of factors (number of hidden layers vs. hidden neurons) to improve the prediction capacity. It is rare to find one single model covering many possible factors or across different performance areas. A comprehensive comparison of the main factors in the ANN structure should be considered in order to find which factors can improve the ANN prediction from different factors.

Based on the Assessment Standard for Green Building [29] and the Assessment Standard for Healthy Building [30], daylight, visual comfort, and outdoor thermal metrics are key considerations during the building design stage. These factors often present conflicting requirements to some extent. Therefore, it is essential to simultaneously consider these metrics when predicting building performance. This study's objective was to find optimized

ANN models to predict the daylight factor (DF), sunlight hours, QuVue (sky view ratio), and universal thermal climate index (UTCI) of residential buildings.

The contribution and novelty of this paper may provide guidance for modeling the ANN to improve the accuracy of building performance prediction.

This study was holistically designed to find the proper way to construct the ANN model for building performance. Some common measures from building simulations were used to see how these different measures can be modeled in an ANN. The sensitivity analysis method was conducted to identify the most important factors to consider in building performance. The objective of the study was to explore the possibility of predicting performance in similar building layouts and create a rule for selecting ANN model parameters that is replicable and widely applicable. For example, multiple performance predictions were carried out for architectural building layout designs that had similar numbers of buildings and exceeded three prediction categories.

## 2. Materials and Methods

This study commenced by employing parameterized modeling of the building layout to identify the independent variables, variables, and constraints. The independent variable was limited to the building volume ratio or total building area. The next step involved parametric performance simulation, which included constructing lighting simulation models, site sunshine models, visual field models, and thermal environment models. This preparation was conducted to collect data for five performance analyses. In the training phase of the ANN model, three types of independent variables from the parameterized model were used as input data: the position coordinates, x-axis and y-axis coordinates of the building unit's relative field, and its height, z. The output data consisted of calculation values from the simulation model, specifically the lighting coefficient, sunshine, vision, and outdoor thermal climate index. By conducting a four-part sensitivity analysis using the ANN architecture, improvements in prediction accuracy were achieved.

Figure 1 summarizes the overall methodology of this study including the random choice of building parameters in building models, simulation of the proposed models in Rhino-Grasshopper (Version 6) software, and training of the ANN models for five building performance estimations.

### 2.1. Input Data Collection

In the cold climate of Beijing, China, the test case involved twelve buildings situated on a site with a total area of 143,358 m$^2$ (as shown in Figure 2). Among these buildings, ten residential buildings (No. 1–6 and No. 8–11) were included in the test, while two commercial buildings (No. 7 and No. 12) were excluded from the calculations (as depicted in Figure 2). It is worth noting that the test did not take into account the urban context surrounding the site due to its location in an open field.

In this study, test data were collected from high-rise residential buildings in Beijing, China. The test buildings included 917 resident units in 10 buildings. Two of the high-rise buildings were for commercial and office use (building numbers 7 and 12). Overall, the residential buildings had 3.1 m of floor-to-floor height and around 0.3–0.4 for the window-to-wall ratio (WWR; Figure 2). More information about the setup for the building simulation models was found at reference [31].

The following section discusses how the training dataset was obtained using different computational simulation tools in detail (Figure 3). In this study, considering the rectangular shape of the buildings, the methodology scope was bounded to residential buildings. The 12 test buildings' spatial position variables (x_n, y_n, and z_n) meant that 36 variables were used (Table 1). The tests used default three-layer ANN models, which were made up of an input layer, a single hidden layer, and an output layer.

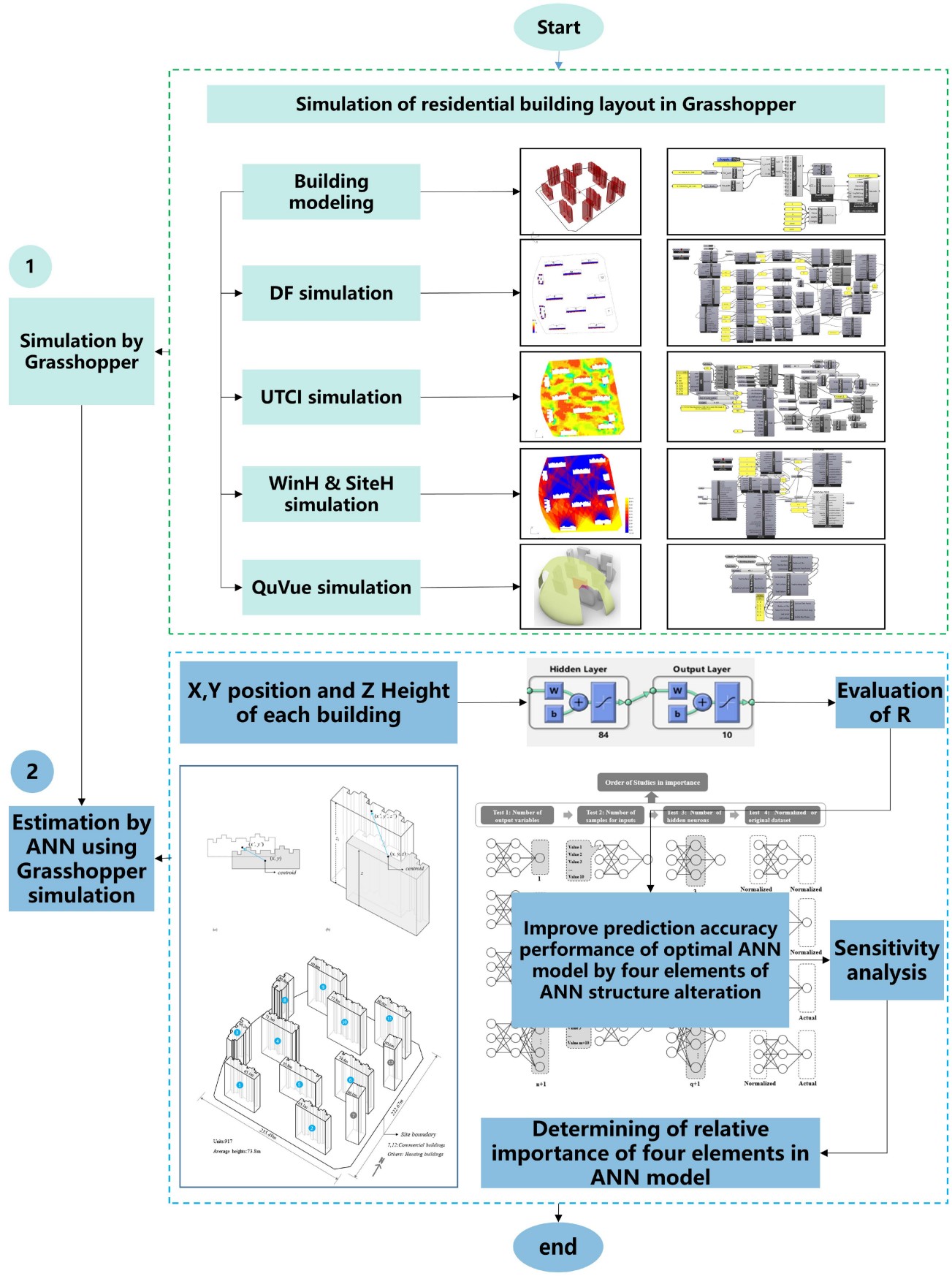

**Figure 1.** Flowchart of this study.

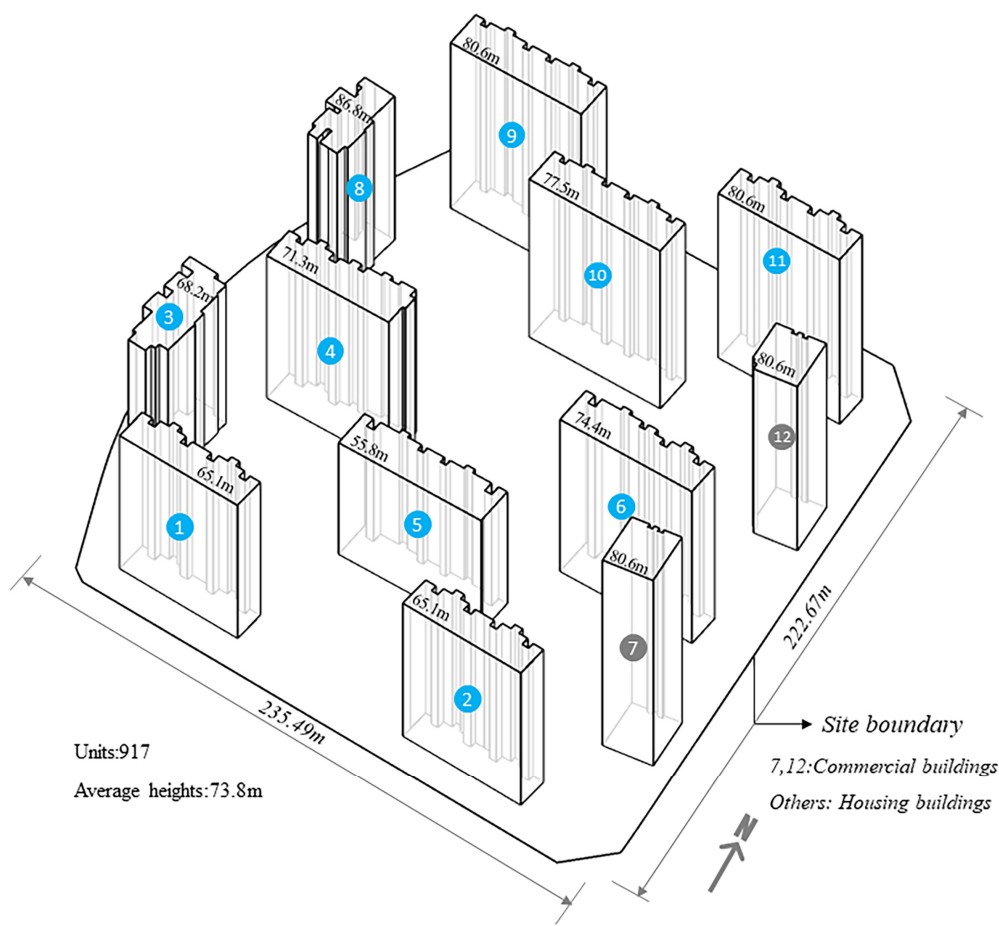

**Figure 2.** Test case building layout.

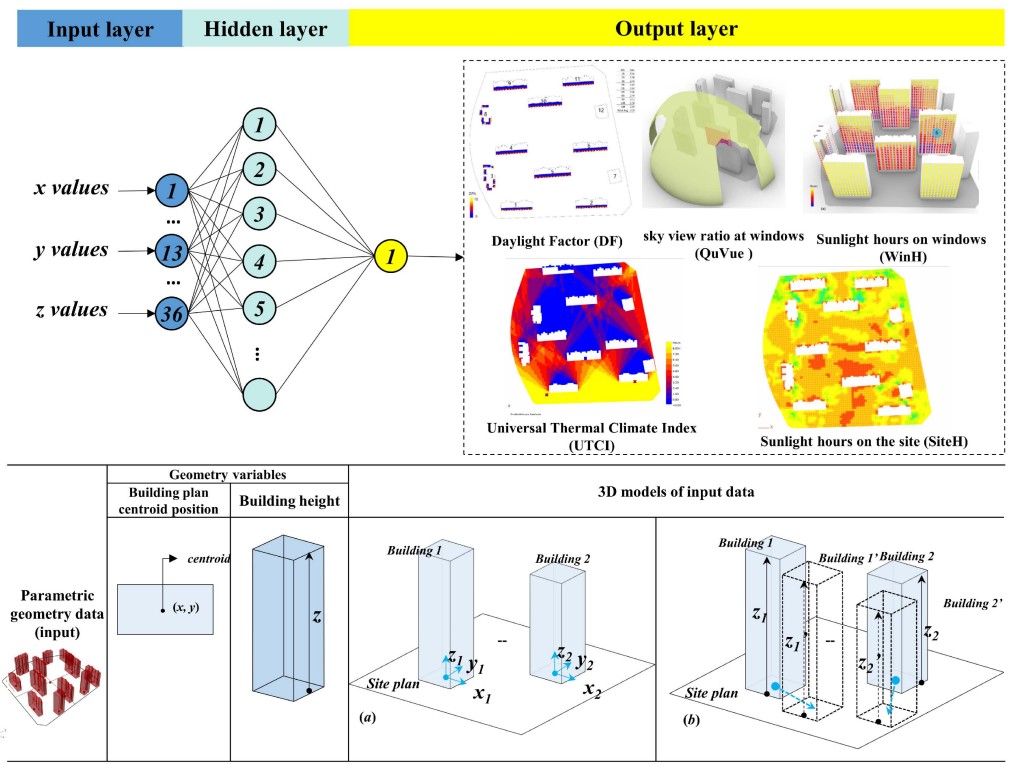

**Figure 3.** Input parameters for ANN model.

**Table 1.** Input and output parameters of ANN models.

| Input and Output Parameters of ANN Models | Quantity | Breakdown |
|---|---|---|
| Input Variables | 36 | x1, x2, x3, . . ., x12<br>y1, y2, y3, . . ., y12<br>z1, z2, z3, . . ., z12 |
| Output Variables | 5 | Daylight factor (DF)<br>Sky view ratio (QuVue)<br>Window sunlight hours (WinH)<br>Site sunlight hours (SiteH)<br>Universal thermal climate index (UTCI) |

The geometry-related variables were used as the input variables and different simulation results were used as the outputs for training the ANN. The controlling geometry was scripted through the non-uniform rational basis spline (NURBS) CAD (computational-aided design) tool that automated the process of updating its geographical configuration. The variables of the input parameters were the spatial position of each building (n), which were symbolized as x_n, y_n, and z_n. The objective of optimizing the design of the residential building layout was to meet the specified design requirements, such as controlling the buildings so as not to intersect with other buildings, buildings not being moved beyond the site boundary, and the total building floor area being between ±10% of the original case. The range of values was determined according to the site status and the Standard for Urban Residential Area Planning and Design [32], the test buildings were able to move along both the x- and y-axes, and the possible moveable range from −10 to 20 m was based on the initial building location. The average height of a floor was 3.1 m. The building height (z) can change from 34.1 to 99.2 m according to the High-Rise Building Code for The Fire Protection Design of Buildings [33].

The generation of random input values in these experiments was governed by the building standard to ensure control. It is important to note that these experiments were solely conducted for obtaining ANN models and not intended for actual design purposes. The input variables were randomly generated by a statistics tool according to the variable range mentioned above.

*2.2. Output Data Collection*

This research applied Rhino-Grasshopper to simulate five performance measures: DF, QuVue, WinH, SiteH, and UTCI. The details of the five stimulation tools with their main settings are shown in Table 2. This section aims to collect data from the results of Rhino-Grasshopper V6. The following section discusses the five measures used as outputs for training the dataset. The five measures were all related to the building indoor and outdoor environmental conditions, such as daylight, view, and thermal conditions for indoors and outdoors. These five measures included the daylight factor (DF), sunlight hours on the window (WinH), QuVue (sky view ratio at windows), sunlight hours on the site (SiteH), and the universal thermal climate index (UTCI). More details on each measurement are discussed below.

The daylight factor (DF) was simulated with Ladybug in Grasshopper, and measurement points were located on the first floors of all 10 test residential buildings. A 1.5 × 1.5 m grid 0.75 m above the floor level was used for calculation, which consisted of 2888 measurement points in total. The average values of the DF of all points were considered as the output values of the DF.

The sunlight hours on the window (WinH) were also calculated by Ladybug. The time duration of sunlight hours was measured from 8:00 a.m. to 16:00 p.m. on January 21st (the coldest day according to the Chinese building code). Each window on the first floor of the 10 test buildings had one average measuring value 0.8 m above the floor.

**Table 2.** Characteristic settings in the simulation.

| Metrics | Simulation Tools | Constant Items | Values |
|---|---|---|---|
| DF | Ladybug (ver. 0.061) (Radiance) | Location and weather file | Beijing |
| | | Grid size | 1 × 1 m |
| | | Distance from base surface | 0.75 m |
| | | Sky | Uniform CIE sky |
| | | Radiance parameters | -ps 8, -pt 0.15, -pj 0.6, -ds 0.5, -dt 0.5, -dc 0.25, -dp 64, -ab 0, -aa 0.15, -ar 32, -as 32, -lr 4, and -lw 0.05 |
| | | Window width-to-height ratio | 1.2/1 |
| WinH | Ladybug (ver. 0.061) | Date and time | Jan 21 8:00–16:00 |
| | | Simulation time steps per hour | 1 |
| | | Grid size | 3 × 3 m |
| SiteH | Ladybug (ver. 0.061) | Site grid size (SiteH) | 2.5 × 2.5 m |
| | | Date and time | Jan 21 8:00–16:00 |
| | | Simulation times step per hour | 1 |
| UTCI | EDDy3D (blueCFD) | Wind direction | 0, 45, 90, 135, 180, 225, 270, and 315° |
| | | Boundary type | cylindrical domain |
| | | Boundary inner rectangle | 400 m |
| | | Boundary outer radius | 1000 m |
| | | Boundary height | 250 m |
| | | Mesh size | 357,568 |
| | | Mesh type | OpenFOAM's blockMesh and snappyHexMesh |
| | | CFD turbulence model | kOmegaSST |
| | | Pressure model | SIMPLE (Semi-implicit method for pressure-linked equations) |
| Sky view ratio | QuVue | Test surface | South/east side windows |
| | | Measuring point | Center of each window |

To calculate how much open sky view each residential unit had, the study used a calculator called QuVue. It can calculate the open sky view more realistically than other measures [12]. It used the same measuring point setup as for the sunlight hours on the window (WinH). The results of the QuVue calculation were the average values of all measuring points.

Sunlight hours on the site (SiteH) were the sunlight duration on the site from 8:00 a.m. to 4:00 p.m. on the coldest day of the location (Jan 21), same as the measuring condition of WinH. The site was meshed with a grid size of 2.5 × 2.5 m. The measuring height was set at 1.5 m from the ground. A total of 4454 measurement points and the average values of points were the output parameter.

The universal thermal climate index (UTCI), consisting of temperature, relative humidity, solar radiation, and wind speed, is an outdoor thermal metric that was used for the test. For the UTCI calculation, the measuring grid size was 1.5 × 1.5 m at the height of 1.5 m from the ground. A total of 7938 measured points were used, and an average value of the points was used for the test. To calculate the UTCI, the study used Eddy3D in Grasshopper, which used a CFD (computational fluid dynamics) engine to calculate the wind speed and direction for the UTCI calculation.

The average values were surveyed because, in building design standards in China, most metric values are considered averages, along with minimum and maximum values. In future studies, other solution methods for values will be given greater attention.

As discussed above, test 4 was to find how the normalizing input and output dataset could improve prediction. As shown in Table 3, the input and output values consisted of a wide range of values that might reduce the accuracy of the ANN model [34,35]. To mitigate this problem, some studies have proposed that all the input and output data could

be normalized using a min–max approach, which checks whether the computation and performance of the ANN models could be enhanced by normalizing all the input and output data ranging from 0 to 1. The min–max normalization equation is shown as follows:

$$\beta\_nor = (\beta\_i - \beta\_min)/(\beta\_max - \beta\_min), \tag{1}$$

where $\beta\_max$ is the maximum value of the attribute and $\beta\_min$ is the minimum value of the attribute.

**Table 3.** Statistical measures for inputs and outputs in building performance.

| Variables | Average | Std. Dev | Minimum | Maximum |
|---|---|---|---|---|
| Inputs | | | | |
| $x_n$ (m) | 3.872 | 8.423 | −10.583 | 19.417 |
| $y_n$ (m) | 18.532 | 12.249 | −3.400 | 39.644 |
| $z_n$ (m) | 44.036 | 20.211 | 9.300 | 80.083 |
| Outputs | | | | |
| SiteH (h) | 0.488 | 0.026 | 0.432 | 0.544 |
| UTCI | 0.574 | 0.177 | 0.074 | 0.935 |
| DF (%) | 6.615 | 12.115 | 3.359 | 72.935 |
| WinH (h) | 4.789 | 0.980 | 2.667 | 6.894 |
| QuVue (%) | 32.448 | 7.981 | 11.538 | 45.527 |

*2.3. Development of ANN Models*

To find the best way to build the ANN model for different building performances, it is important to conduct a comparative study of the ANN models with different built environment measures to gain general insight into the relationship between the prediction results, the training model's setup, and the dataset. The ANN model's performance was measured by the correlation coefficient (R) between the predicted data and actual data to evaluate the performance of the established ANN models.

As shown in Figure 4, the variables of the input layers were represented as $a_n$ and the output layer variables were represented as $b_n$. Each variable contained the same number of sample sizes that had different values (Value$_n$). The hidden layer consisted of "n" numbers of neurons, represented as $c_n$.

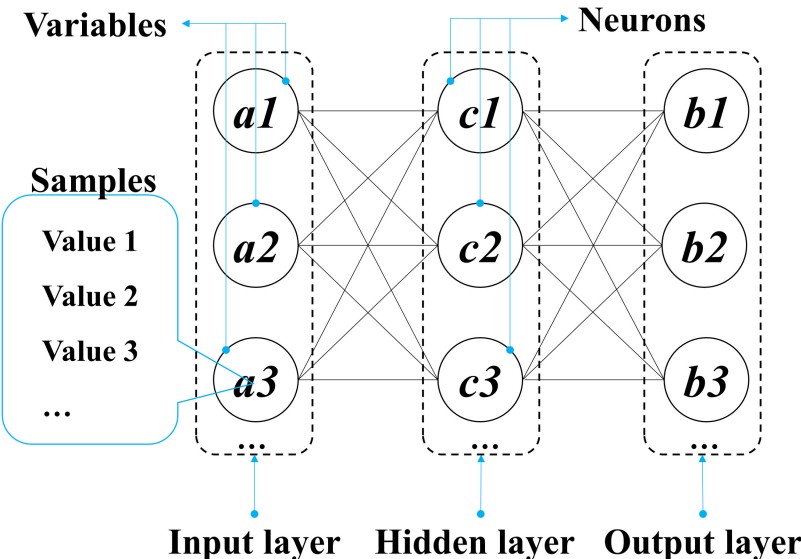

**Figure 4.** The definition of the standard structure of an ANN model.

Figure 5 demonstrates the four tests discussed above. Test 1 showed a different number of output layer variables ($b_n$). In test 2, the different numbers of sample sizes ($Value_m$) were tested to discover the difference between the number of samples and accuracy. For test 3, a q number of neurons was used for the hidden layer neurons ($c_q$). In test 4, four ANN models of different input and output variables (normalized vs. normalized, actual vs. normalized, actual vs. actual, and normalized vs. actual) were constructed.

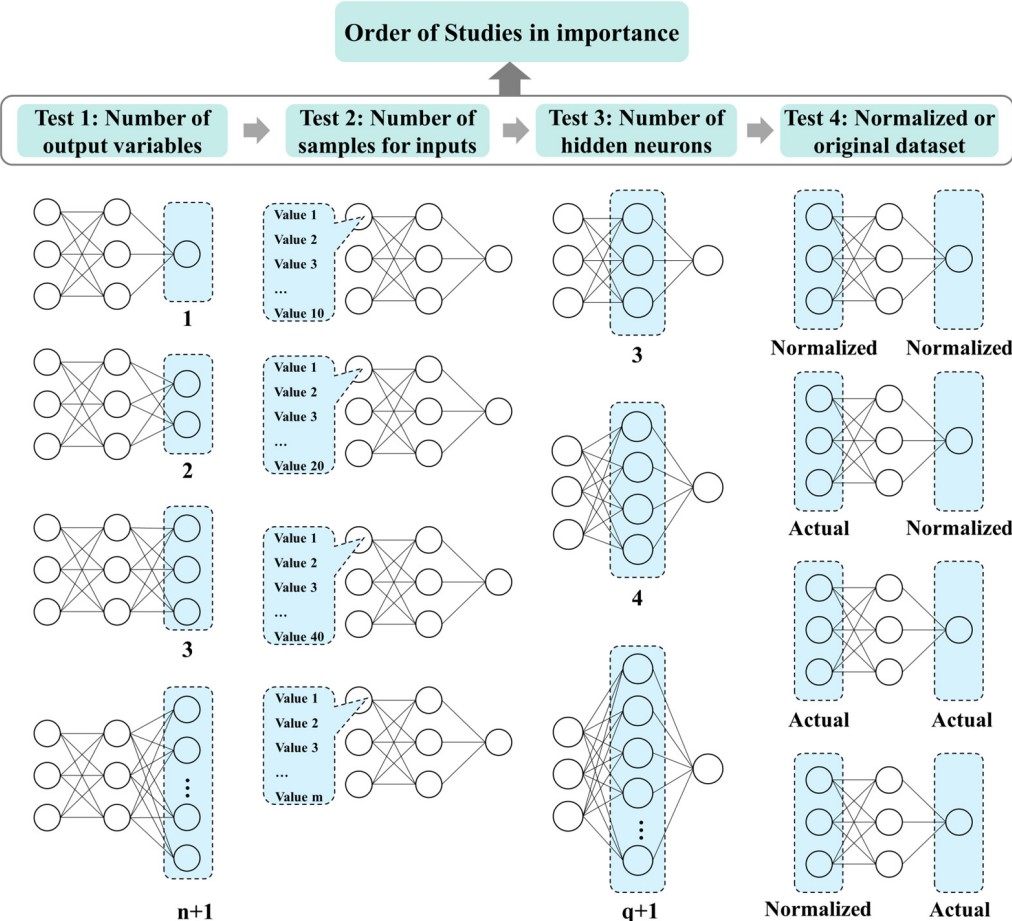

**Figure 5.** Overall view of neural network sensitivity analysis.

The procedure of using ANNs to perform prediction and the evaluation of the performance of the ANNs involved the four tests are as follows:

Test 1: number of output variables. This test was designed to question how many training output variables were reasonable with a fixed number of training input variables. It is common in building performance measures that several different measures are used to evaluate performance. For instance, in LEED v4 daylight, sDA (spatial daylight autonomy) and ASE (annual sunlight exposure) are needed, and both are related to the same input variables such as the window size and location. What is the ideal method to build an ANN model that has the same input variables? Would it be better to build two measures in one ANN model or two separate models? Test 1 was designed to understand this question of building an independent ANN model for each measure, or the possibility of combining them together. The purpose of this test was to improve the prediction efficiency if one ANN model has the capacity for higher prediction accuracy with more output values.

Test 2: number of samples for inputs. This test was designed to understand whether having more training input cases would increase the accuracy of the ANN models. This purely depends on the complexity of the problem and commonly requires more cases to improve accuracy. However, there is a question of what number of cases is good

enough. The answer depends on the other elements such as the number of output and input variables. Escandón and others indicate that a ratio of 2.5 between the sample size and the number of characteristic parameters (input variables) would be a reasonable sample case to train a model [24]. The proposed question was whether this relation between the input variables and the number of training datasets would be linear. Would there be a point where the linear curve line would become a plateau? The goal of this test was to use proper quantities of samples to lead the optimal ANN models in predicting performance.

Test 3: number of hidden neurons. It is known that there is a range of ratios between the number of input variables and the number of hidden neurons that can develop accuracy. As in test 2, would a greater number of hidden neurons increase the accuracy, or would there be a point that increasing the number of hidden neurons would not further improve the accuracy? This test was intended to find some rules showing how many neurons in the hidden layers are necessary to reach high ANN model accuracy.

Test 4: normalized or original datasets. This test compared the combination of normalization and actual input and output values to decide which pair of groups achieve better accuracy. The normalization of variables was applied in a wide range of values, especially in different orders of magnitude. However, the difference in the ANN model accuracy of normalized or actual values revealed the significance of the normalization of the dataset. Upon reviewing the version of MATLAB utilized in our research, we found that the normalization step was not included in the procedure. For each test, the training process was repeated 30 times for all ANN models, and the average value was employed to facilitate comparison among different types of datasets for result validation purposes. This test aimed to discover the rules of the normalization of datasets to achieve optimal ANN models in terms of prediction accuracy ability.

This study hoped to gain certain insight into the stability and performance of ANN models through the four tests discussed above. The selection of every test led to a comparison in the end, which yielded a summary and principles to show the characteristics of how the importance of each element influenced the prediction of the ANN models.

In this research, ANN-MLP models were used to train the ANN models. For the ANN model, a three-layer feedforward network with sigmoid hidden neurons and linear output neurons that could fit multi-dimensional problems was adopted. The transfer function was a hyperbolic tangent function in the hidden neurons, and a linear function was used in the output neurons.

For training the ANN model, MATLAB's Deep Learning Toolbox was used for the test. To find the overall performance in terms of accuracy, this study conducted 30 different trainings for each test, which allowed us to reach more stable correlation coefficients R. In the ANN calculating procedure, there were four R values for testing, training, validation and all data. In order to find network performance, the average of all the R data is applied in this paper. Basic ANN model setups were employed and revised to reflect the different configurations for different tests.

A base ANN model (base case) setup was used to perform the comparison work among the four tests, and the following setups for the ANN model were used for this study (Table 4). Because the test case had 12 buildings (n = 12; Figure 3), a total of 36 variables, including from x_1 to x_12, y_1 to y_12, and z_1 to z_12, were used as the input variables. As discussed in Section 2.2, the performance metrics DF, WinH, QuVue, SiteH, and UTCI were the five output parameters. For the sample size of the ANN, we randomly selected 52 samples, which was about 1.5 times the number of input variables. The number of hidden neurons was set to 108, which was about 3 times the number of input variables. More detailed information is discussed in Table 2. The values of inputs were normalized to the interval (0, 1).

**Table 4.** Base ANN model setup.

| ANN | No. of Inputs | Number of Neurons | Layer | Training Function | Transfer Function | |
|---|---|---|---|---|---|---|
| | | | | | Hidden Neurons | Output Neuron |
| | 36 | 108 | 3 | Levenberg–Marquardt backpropagation algorithm (trainlm) | Hyperbolic tangent function | Linear function |
| Data Division | Training: 70% of dataset | | | | | |
| | Simulation: 15% of dataset | | | | | |
| | Validation: 15% of dataset | | | | | |

### 2.3.1. Number of Output Variables

The first test investigated the accuracy with a number of output variables. It is common to have one output variable as a training dataset. However, sometimes it is beneficial to have more than one output variable as a training dataset. If the outputs are similar measures that use the same input variables, then it would be beneficial to combine two independent ANN models into one. For instance, in daylight simulation, sDA and ASE use the same input parameters to calculate the measures, and if one ANN model can predict both measures, then it would be beneficial to reduce the computational time and power. For this reason, this study developed three models to test their accuracy with a different number of outputs.

Three models (models A, B, and C) were built with the same inputs for building the geometry parameters and a different number of output variables. The comparison of models A, B, and C by different assemble approaches provides insight into how to increase the accuracy of the predictions. The ensemble approach was based on the idea that by combining similar forecasters, it would be possible to improve the overall forecasting accuracy, which was used to improve performance models in another study [36].

All three models had the following ANN setups to keep more parts of the structure of the ANN models accordant. A two-layer feedforward network with sigmoid hidden neurons and linear output neurons that could fit multi-dimensional problems was adopted. It also had the same 36 input variables and 108 neurons in one hidden layer (3 times the number of input variables). The only difference was the number of output variables.

Model A had five independent ANN models for five different outputs (DF, WinH, QuVue, SiteH, and UTCI). Figure A1 shows the ANN structure used for model A. Model B grouped the output variables into two with related measures. One group was composed of indoor measures including DF, WinH, and QuVue (Figure A2). Another group of measures included the outdoor conditions of SiteH and UTCI (Figure A3). Model C included all five output variables in one model as shown in Figure A4. It used one model to train five outputs including indoor measures of DF, WinH, QuVue, and the outdoor conditions of SiteH and UTCI.

Each model conducted 30 trainings and the R values of the forecasting models were analyzed by R distribution analysis as shown in Figures 6–8. From the R distribution histograms, the 95% confidence interval of the error distributions is located in different ranges for models A, B and C. The average values ($\mu$) and standard deviations ($\sigma$) are listed in each histogram. All of the standard deviation values are smaller than 0.12, meaning a low degree of dispersion, and the average R can be used to express the average accuracy performance of each model.

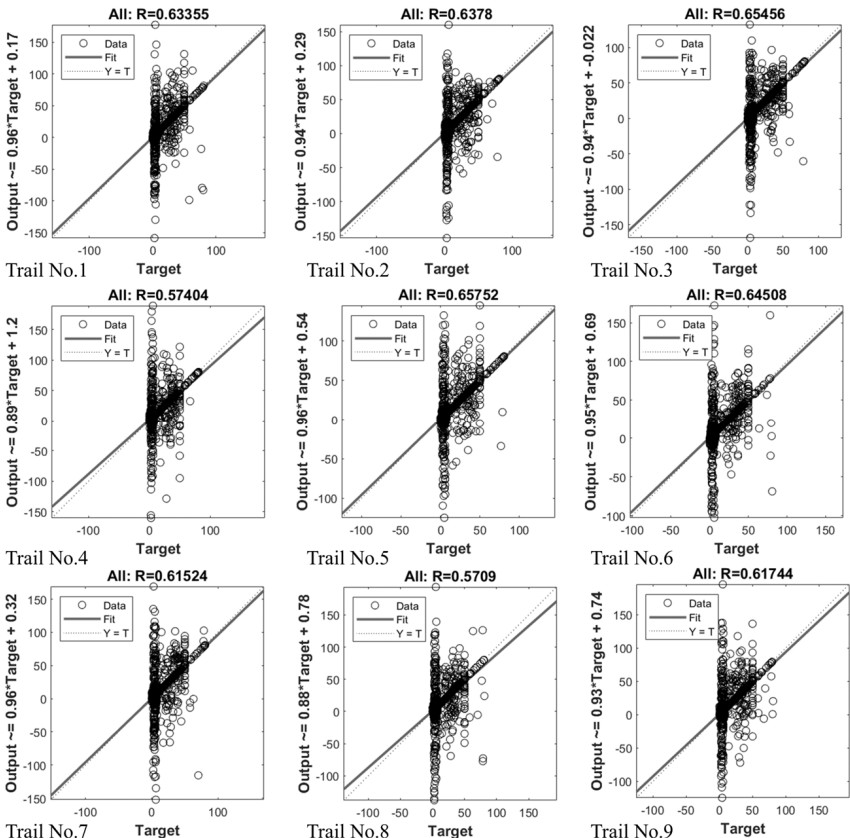

**Figure 6.** R value alteration of trials 1–9 in model A.

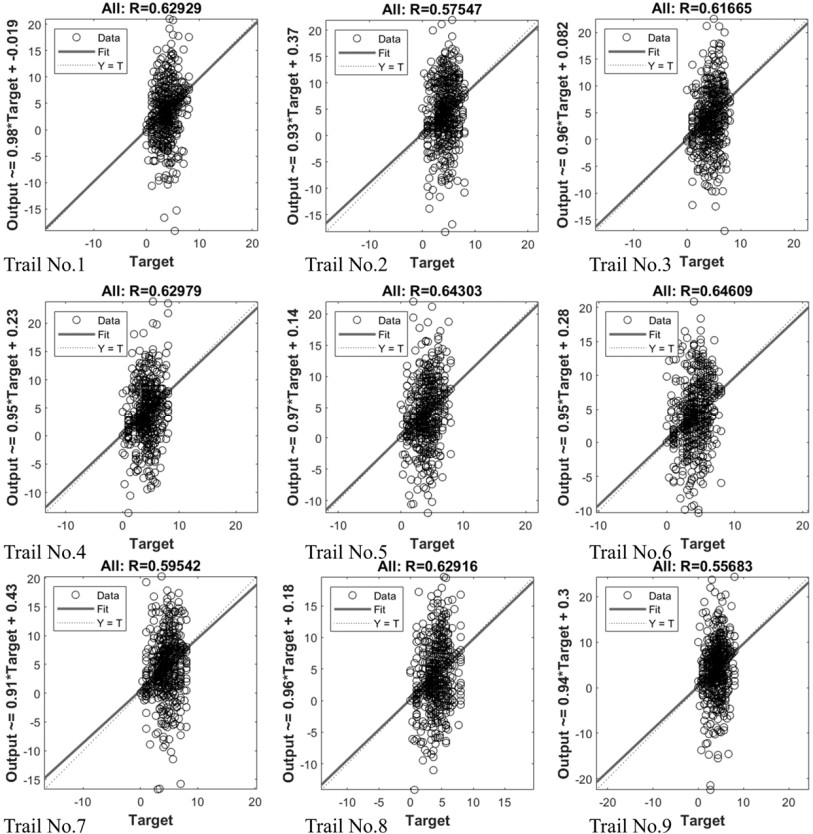

**Figure 7.** R value alteration of trials 1–9 in model B.

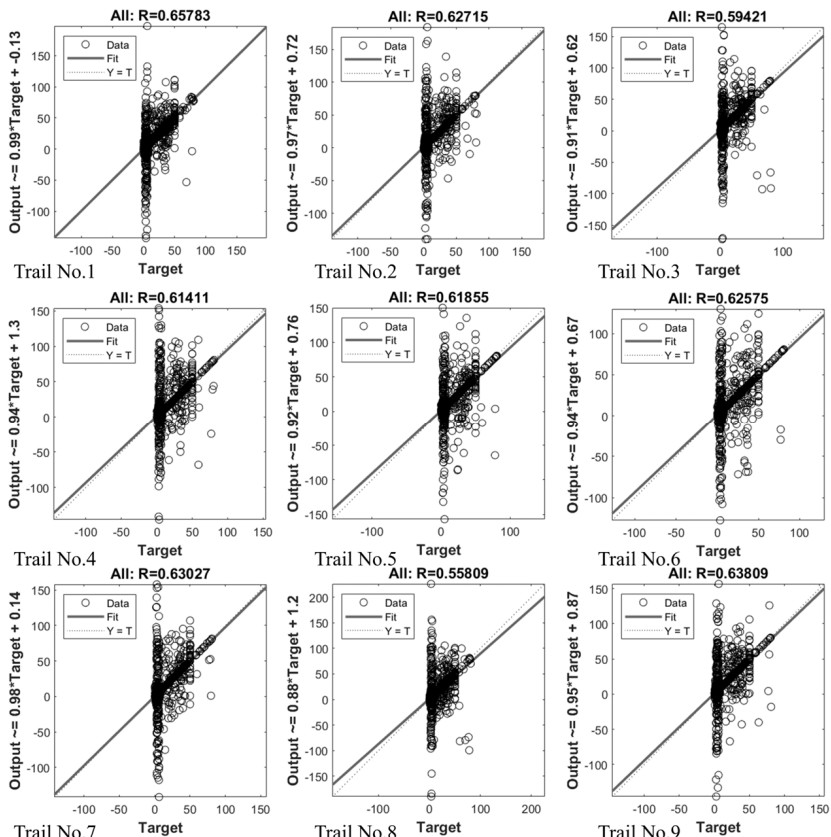

**Figure 8.** R value alteration of trials 1–9 in model C.

### 2.3.2. Number of Training Samples

It is commonly acknowledged that the number of samples is considered a critical factor impacting ANN performance [37]. This test was designed to understand the correlation between the number of samples and its accuracy. The test conducted three different models with output training datasets that included DF, QuVue, and WinH. To eliminate influence from other parameters, all three models used the same training functions as mentioned above with 108 hidden neurons and one hidden layer.

The number of input samples used for the test were 20, 30, 40, 52, 72, and 108. Based on the input variables (36), as a reference, a sample size 2 to 3 times the number of variables was recommended for the test, which was around 90 samples. Each test applied 30 independent trainings to find the R values. As shown in Tables A1–A3, 30 trials were conducted to reach a stable accuracy of the R value.

### 2.3.3. Number of Hidden Layer Neurons

The number of hidden layer neurons depends on the ANN model's complexity. A redundant number of neurons in hidden layers will lead to overfitting the ANN model and take more time for training. The numbers of hidden layer neurons used for the test were 36, 54, 72, 90, 108, and 144, which were 1, 1.5, 2, 2.5, 3, and 4 times the number of input variables (36).

### 2.3.4. Normalized or Original Datasets

The normalization of the dataset in ANN models has been reported in previous studies to have stability and efficiency in creating a better performing neural network. However, for some cases, the normalized dataset has little significant influence on the accuracy of the neural network models [22]. To find whether normalization would improve the prediction

of building performance or not, the normalized and raw datasets were considered as both inputs and outputs.

Like in tests 2 and 3, the base ANN model with a sample size of 52, 108 neurons in the hidden layer were used. The test investigated three different models that contained different output datasets including DF, QuVue, and WinH. The input dataset for all three models used the same 36 variables.

These three models' input and output training datasets were modified to have four different training datasets. Dataset 1 normalized both the input and output datasets, dataset 2 normalized the input but not the output, dataset 3 did not normalize the input but normalized the output, and dataset 4 did not normalize datasets for either the input or output. As in the previous three tests, training was conducted 30 times for all ANN models, and the average value was used to compare the different types of datasets.

### *2.4. Performance Evaluation*

The correlation coefficient R, a regression index, was used to evaluate the performance of the ANN, which measures the correlation between the predicted and actual value [38,39]. R values vary between −1 and +1; however, if an R value is closer to 1, then a more positive linear relationship and a high network performance can be obtained. Generally, the root-mean-squared error (RMSE), mean-squared error (MSE), mean absolute percentage error (MAPE), coefficient of determination ($R^2$), and MAE are commonly used performance metrics of ANN models together with the regression R [40–42]. The MSE is the average squared difference between the ANN predictions and performance simulations. The MSE is used to calculate the average squared difference between the estimated value and the simulated value, lower values of which indicate that the data fit better. The training process can automatically stop when the mean-squared error (MSE) of the validation samples is stabilized. Escandón et al. used a regression analysis with a coefficient of regression (R) and relative errors to show the reliability of the developed ANN model in predicting the energy performance and thermal comfort of a social housing stock in southern Europe [24], similar to previous studies related to a building stock [43]. Therefore, in this paper, we also used the R value to determine the performance of the ANN models [44,45].

### 3. Results

### *3.1. The Results of the Four Developed ANN Models*

We explored the possibilities of the performance prediction problem of similar building layouts and provided a replicable and widely used ANN model parameter selection rule. For example, under a similar number of complex architectural layouts, more than three kinds of performance predictions are made. For different indicators, the independent variables were selected based on Tables 2 and 3.

#### 3.1.1. Number of Input and Output Variables' Results

The frequency of distribution of x_n, y_n, and z_n variables was displayed in Figure 9. The frequency of the x_n and y_n variable values were between 0.1 and 0.2, except for the value range between −15 and −10 m (Figure 9, left). As can be seen in Figure 9 (right), the z_n variables were randomly grouped into two, one group in the 0.1–0.14 frequency range and another group in a frequency between 0.04 and 0.08.

Figure 10 describes the ratio distribution achieved in the simulation results of five output metrics, which were WinH, DF, QuVue, UTCI, and SiteH. The highest frequency for the WinH simulation result ranged from 2 to 4 h. The simulation results between 0 and 2 h had the lowest frequency value, 0.10. For the DF simulation results, the maximum frequency was 0.45 when the DF value was between 0 and 4%. The lowest frequency was when the DF value was between 4 and 6%. For QuVue, the number of frequencies increased considerably from 0.05 to 0.30 when the QuVue result values were between 0 and 40%. The range of QuVue results from 40% to 50% had about the same frequency ratio as the range from 30% to 40%. The range of UTCI result values was from

0.8 to 1.0, in which the highest frequency was recorded at 0.45. Then, the frequency from 0.2 to 0.4 in UTCI ranked second, at 0.4. The frequencies of 46–49% and 49–52% in SiteH simulation outcomes were similar, exactly 0.35 for each, while in 0–46% and 52–55%, the numbers were considerably lower, averaging between 0.1 and 0.15.

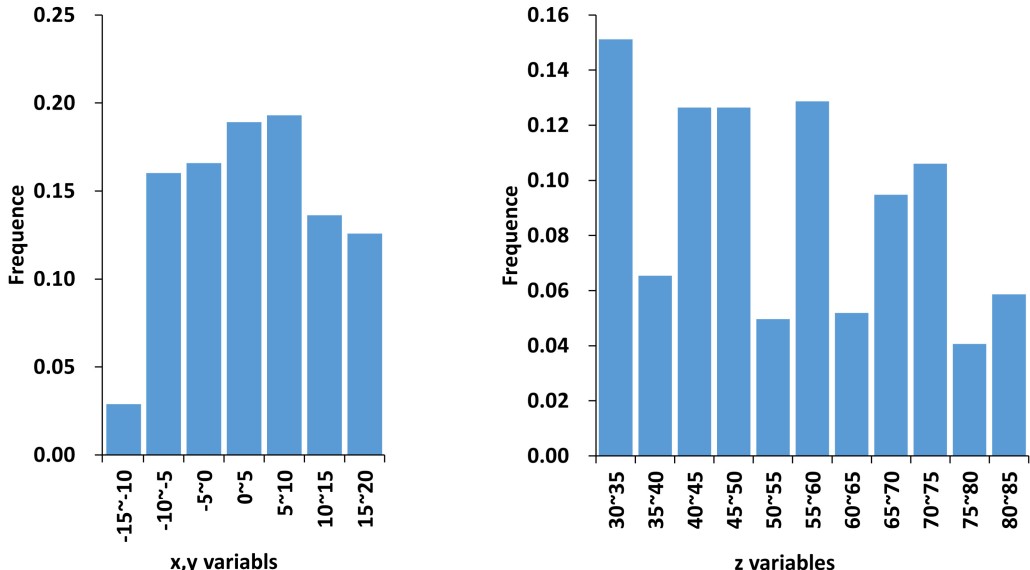

**Figure 9.** Histogram of values of input variables.

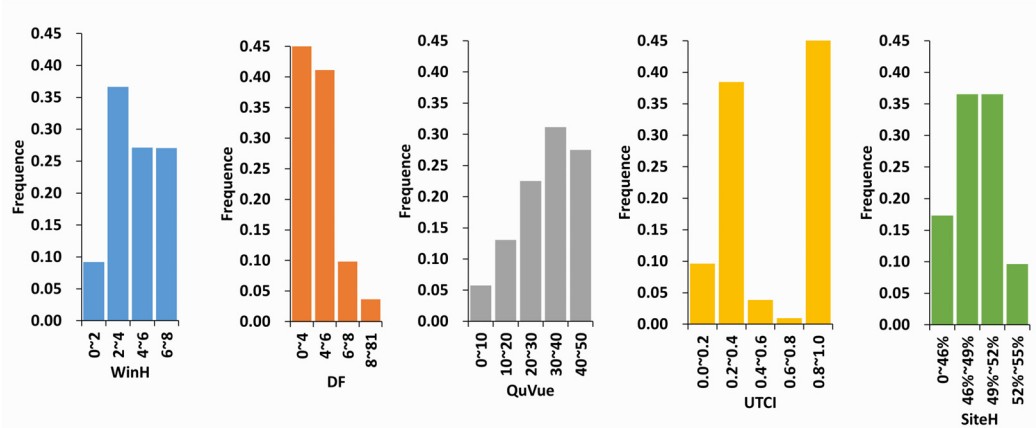

**Figure 10.** Frequency of output values from the five simulation results.

Figure 11 and Table 5 show the R value of the average of 30 trainings for all datasets, which considers the testing datasets and validation datasets. For model A, the R value for each of the five models was 0.226, 0.518, 0.623, 0.323, and 0.503. The average R value of all five ANN models was 0.438. For model B, the R value was 0.605 for group 1 and 0.507 for group 2. The average R value of the two ANN models was 0.556. Model C, which included all five output variables, had an R value of 0.621. The result was that model C made a better prediction than the other two models. Despite the fact that the absolute R value was typically low in most test scenarios, the patterns of variable transformation remained dependable. As a result, greater weight should be given to the relative R value.

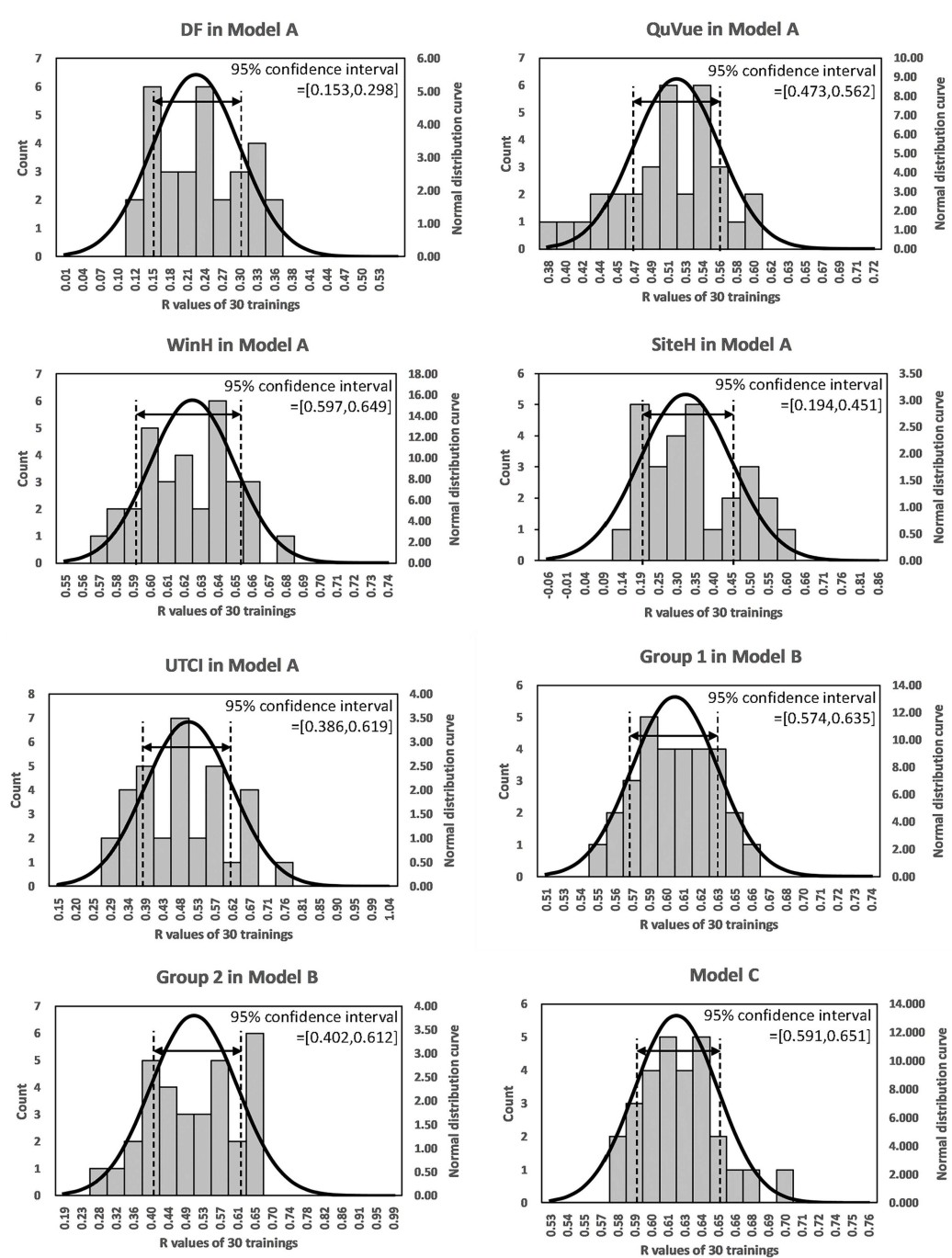

**Figure 11.** R distribution analysis for three models.

**Table 5.** R value of three models.

| | Model A | | | | | Model B | | Model C |
|---|---|---|---|---|---|---|---|---|
| | **DF** | **QuVue** | **WinH** | **SiteH** | **UTCI** | **Group 1 (DF, QuVue, and WinH)** | **Group 2 (SiteH and UTCI)** | **Model C** |
| R | 0.226 | 0.518 | 0.623 | 0.323 | 0.503 | 0.605 | 0.507 | 0.621 |
| Avg. R | | | 0.438 | | | | 0.556 | 0.621 |

### 3.1.2. Number of Training Samples' Results

As expected, all three tests showed an increase in the R value as more samples were used. Figure 12a shows the scatter distribution of the DF ANN model, Figure 12b represents

the scatter plot of the QuVue ANN model, and Figure 12c shows the plot of the WinH ANN model. When the sample size was in the range of 20 to 52, the DF ANN model's R value was increased by 0.0038, which was almost flat. QuVue and WinH showed increases of 0.1314 and 0.157, respectively. When the sample size increased to 70, the comparative increase in the R value between the sample sizes of 52 and 70 was significant. For the DF, it increased by 0.565, which was more than 100 times the increase observed in the sample size of 20 to 55.

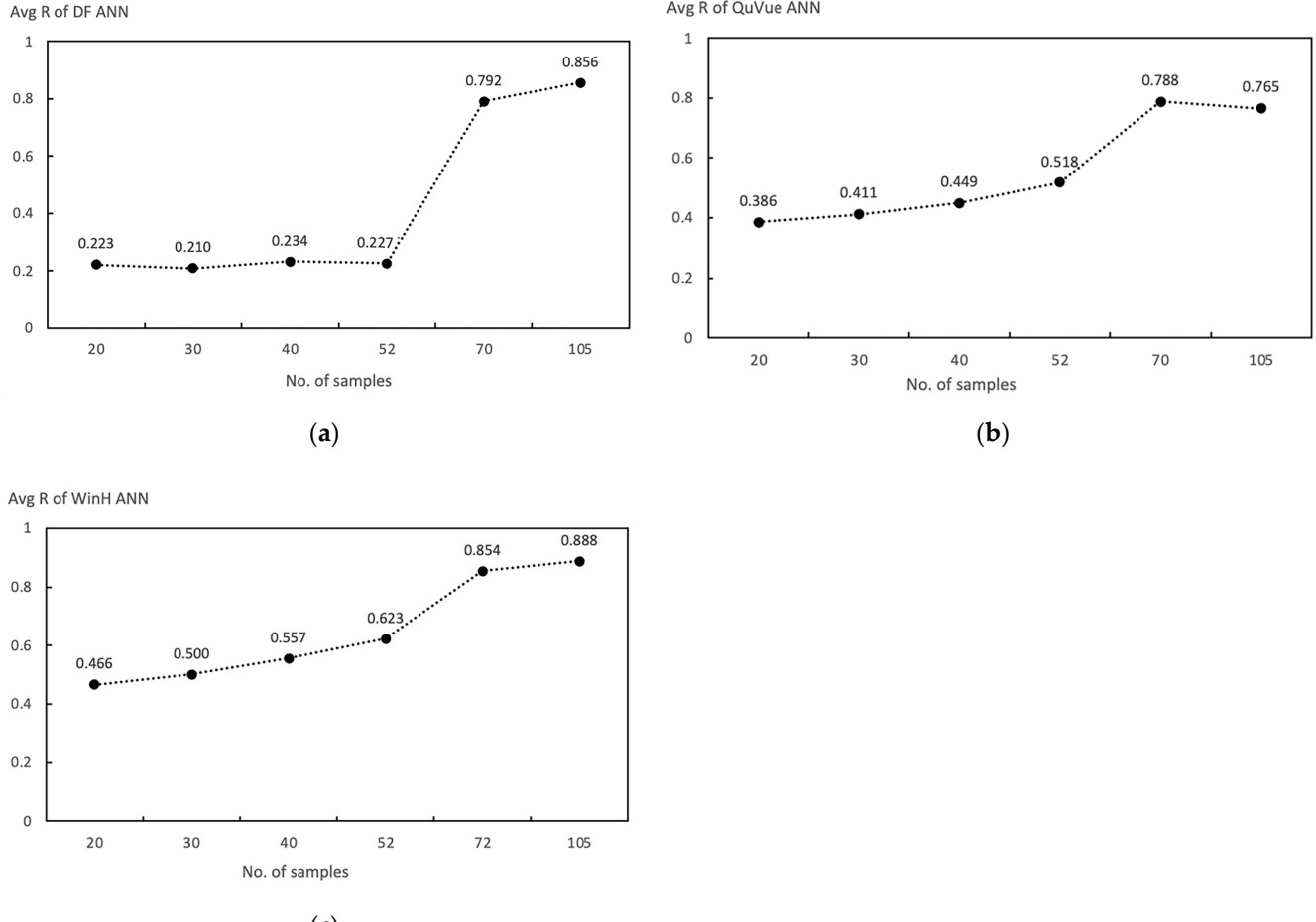

**Figure 12.** Plot of R value and sample size for (**a**) DF ANN model, (**b**) QuVue ANN model, and (**c**) WinH ANN model.

For QuVue and WinH, the increases were 0.271 and 0.231, which were 2.06 and 1.47 times that of the sample size of 20 to 52. However, when the sample size increased to 108, the increase of the R value was not as high as the increase in the case of 70 samples. The DF model was improved by 0.0637 from the sample size of 70, and for QuVue, it was decreased by 0.0233 from the sample size of 70. For WinH, it was increased by 0.0341. It was interesting to see an overserve between the sample size and the accuracy, which showed a nonlinear relation.

Generally, the DF, QuVue, and WinH ANN prediction models showed a similar trend in their R values with respect to the number of training samples. When the number of samples was below 52, the prediction accuracy was low, but it increased significantly once it reached 70, and could achieve an accuracy of 0.8 with an appropriate sample size. As the sample size continued to increase, the level of R improvement becomes limited. Therefore, the ANN prediction model can achieve efficient and effective results as long as the sample size is appropriate.

### 3.1.3. Number of Hidden Layer Neurons' Results

The test constructed three different models with an output training dataset that included DF, QuVue, and WinH. To eliminate the influence from other parameters, all three models used the same training function, number of samples, and number of hidden layers. As in test 2, each ANN model conducted 30 independent trainings to find the stable R values as shown in Tables A4–A6.

The results showed interesting outcomes. For the DF ANN model (Figure 13a), as the number of hidden neurons increased from 36 to 108, the R value of the ANN models decreased by 0.1835. Then, the R value increased by 0.0328 when the number of hidden neurons increased from 108 to 144. However, when the number of neurons was 36, the R value was 0.1507 higher than when the number of hidden neurons was 144.

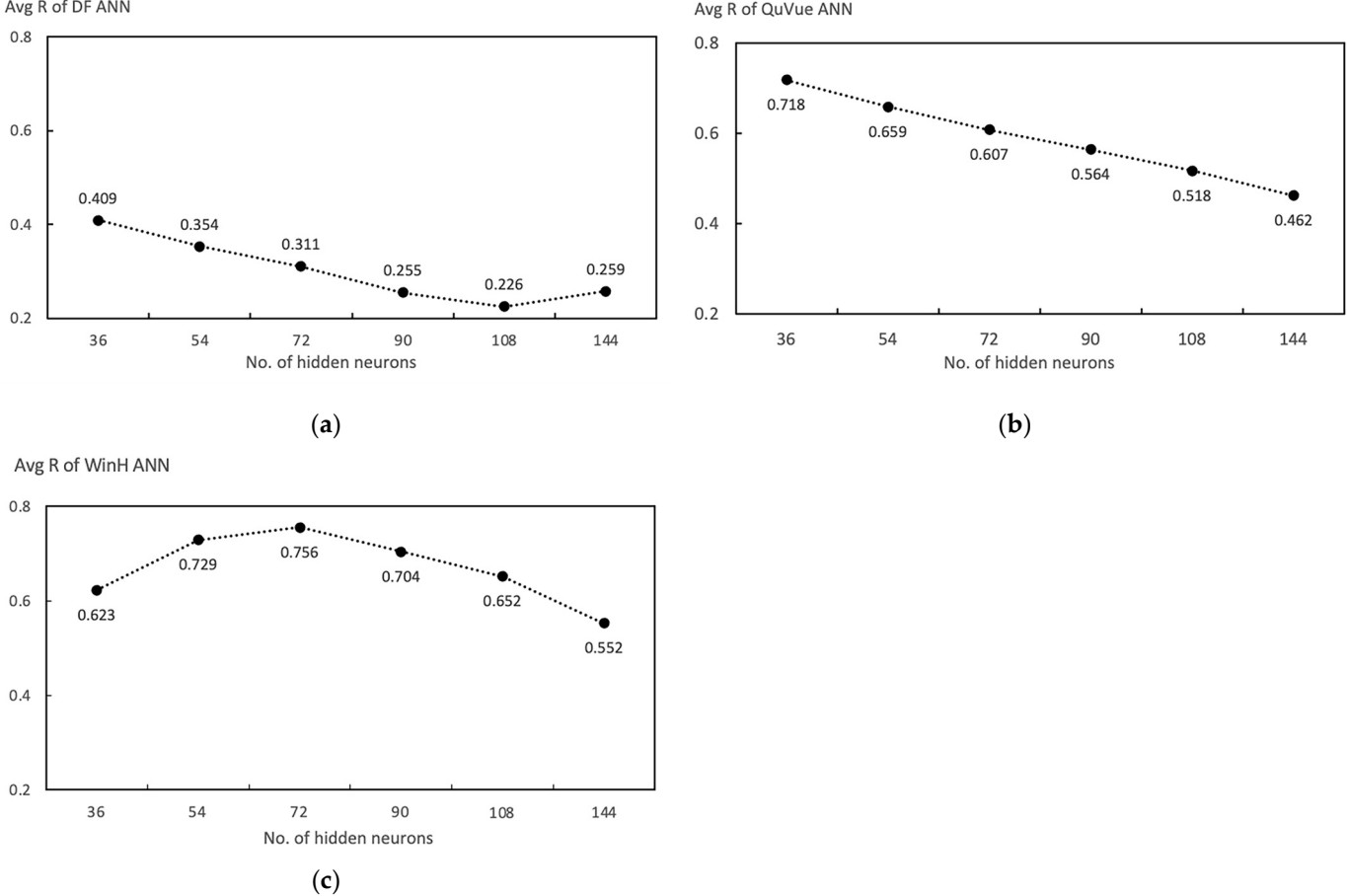

**Figure 13.** Plot of R value and number of hidden neurons for (**a**) DF ANN model, (**b**) QuVue ANN model, and (**c**) WinH ANN model.

The QuVue ANN model (Figure 13b) also had decreasing R values as the number of hidden neurons increased. When the number of hidden neurons was 36, the average QuVue ANN model's R value was 0.7175, and it decreased to 0.4622 when the number of hidden neurons was 144. The average R value in the WinH ANN model (Figure 13c) was 0.7286 when number of hidden neurons was 36, and it decreased to 0.5524 with 144 neurons.

In all three cases, with more neurons in the hidden layer, the R value decreased, except in the DF ANN model when the number of neurons was 144, where the R value was improved compared to when the number of neurons was 108. However, it was interesting to observe that when the number of neurons in the hidden layer was the same as the number of input variables, the models could have higher R values than the recommended hidden number of neurons. Figure 13a,b exhibits a distinct pattern, indicating that the model achieved its highest accuracy when the number of neurons was set to 36. This

observation implies that as the number of neurons increased, there was a noticeable decline in the model's prediction accuracy.

In general, the R values of the DF, QuVue, and WinH ANN prediction models exhibited distinct trends in relation to the number of hidden layer neurons. Both the DF and WinH ANN models demonstrated inflection points in their R values. As the number of hidden layer neurons increased, the R value of the DF and QuVue ANN models exhibited a downward trend, with QuVue showing a linear decline. On the other hand, the R value of the WinH ANN model initially rose and then declined. In the range discussed in this study, only the WinH ANN model identified a more suitable number of hidden neurons, estimated to be less than 36.

### 3.1.4. Normalized Dataset Results

The average and standard deviation of the R value of the four cases in the DF ANN model did not show a difference. Datasets 1 and 4 had a similar R value of 0.2467 and 0.2472, and datasets 2 and 3 had an R value of 0.2257 and 0.2282. Interestingly, normalizing both the input and output and not normalizing either did not show a significant difference. Also, having a combination of normalizing and not normalizing the dataset did not perform better than the other two dataset options. Overall, the standard deviations among the four different options were not significant enough to find better dataset options among the four cases (Figure 14a).

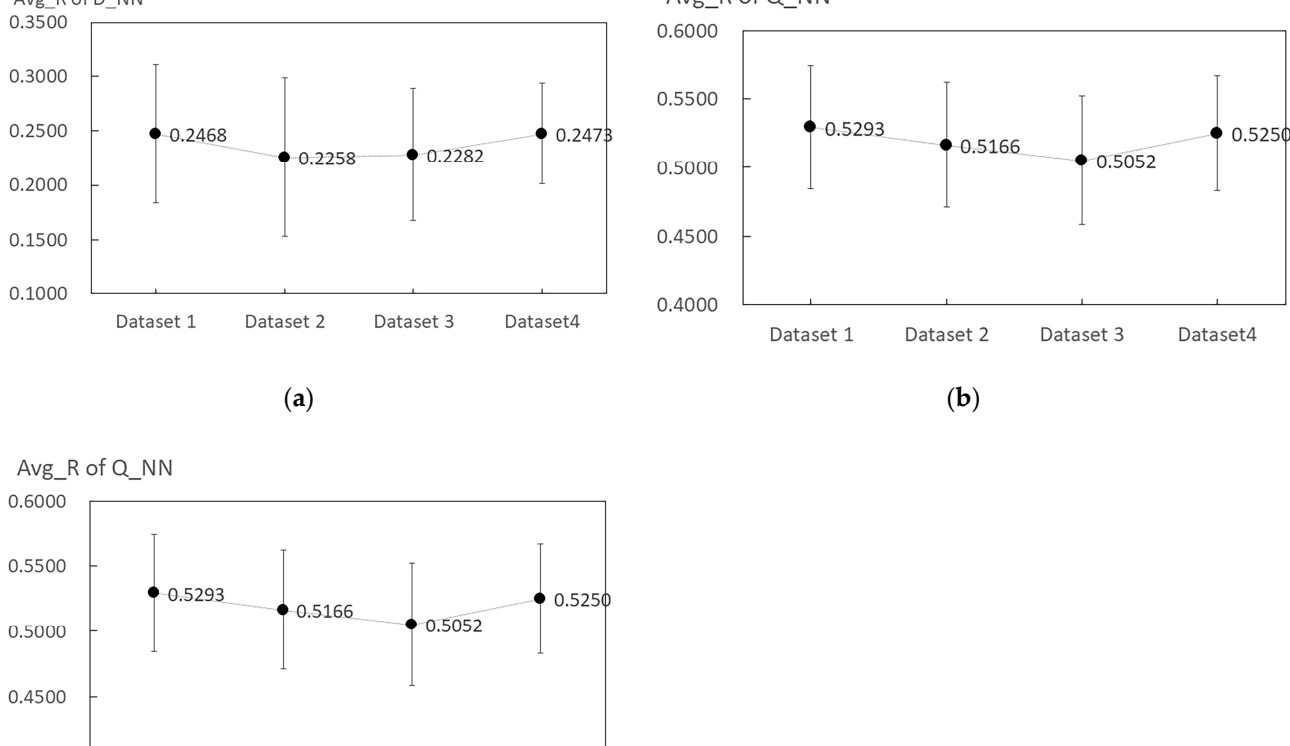

**Figure 14.** Average R value between different dataset configurations for (**a**) DF, (**b**) WinH, and (**c**) QuVue.

Figure 14b shows the average R values of the four cases within the standard deviation, which shows that it was difficult to make one case perform better than the other cases. Similar to the DF model, datasets 1 and 4 had a higher R value than datasets 2 and 3 in the QuVue ANN model. Dataset 1 was 0.529, and dataset 4 was 0.525, which indicates

no significant difference between the normalized dataset and non-normalized dataset. As in Figure 14c, all the average values of the four cases were within the standard deviation, which means that not one option performed better than other dataset options. Overall, from the three models, all three results showed that there was no significant difference between the normalized dataset and non-normalized dataset.

### 3.2. Test Results Analysis

In this section, all test results were compared to find the overall influence of the performance of different ANN configurations on their prediction accuracy. For the relative comparison, a base case ANN model was used to compare the different tests, which had 52 samples for the input dataset and 108 neurons in the hidden layer, with 36 input variables being normalized and the output variables not being normalized.

Based on the base case, the percentage of the altering value of each different ANN model's R value was compared to each other (Figure 15). Figure 15a shows the summary of test 2 of the different number of samples sizes' impact on accuracy. In test 2, three ANN models were tested, which had the same input variables but different output variables. The figure shows the average R values of three models with a different sample size. It can be seen in Figure 15a that having a greater number of samples in the inputs can enhance the R value. However, the trend line was not linear after the sample size increased to more than 72, and the slope was not rising sharply. Compared to the base case (52 samples), the R value of the sample size of 72 increased significantly from 0.62 to 0.85. Based on the test, around two times the number of variables was a reasonable range to select as the sample size.

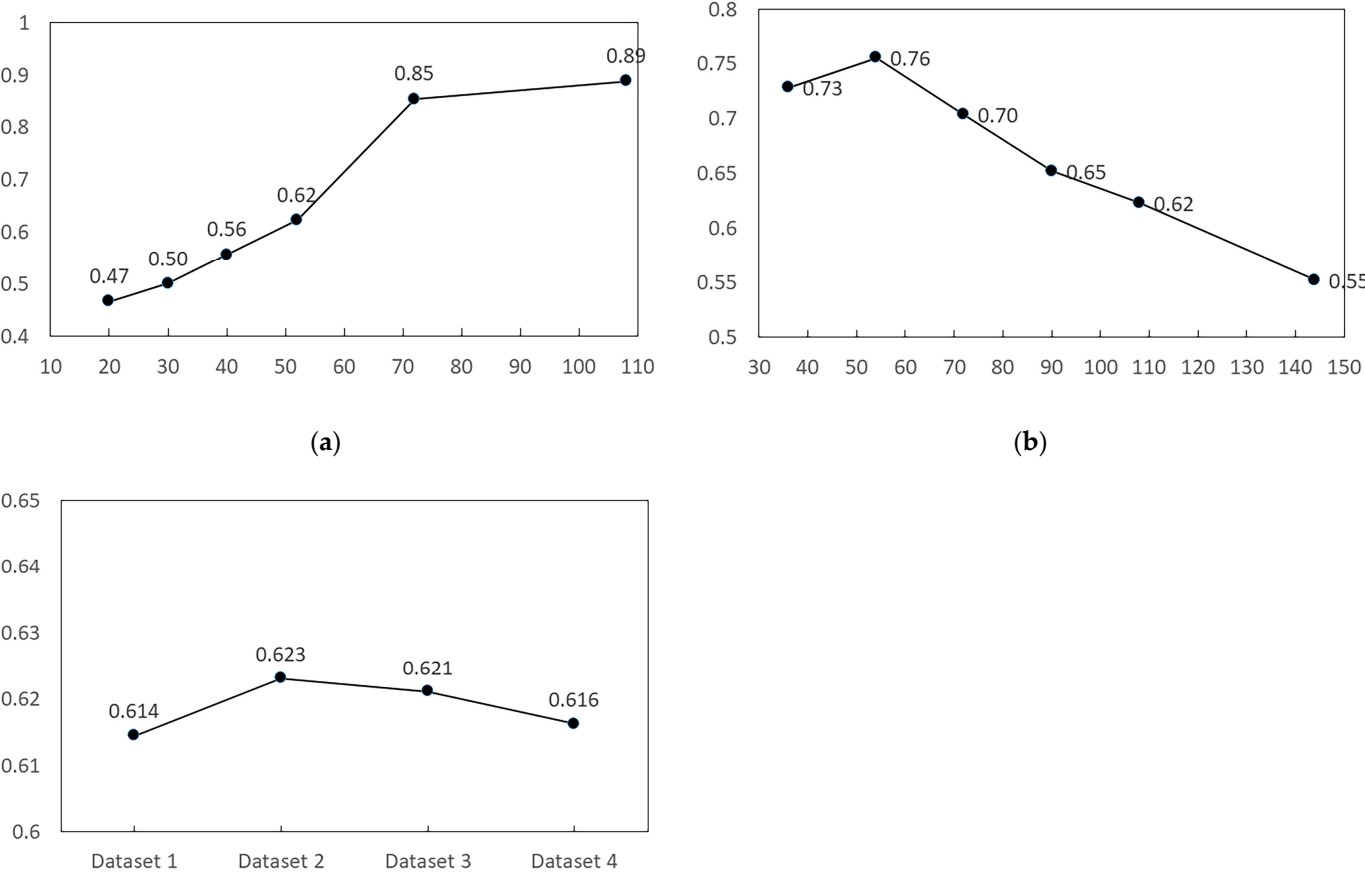

**Figure 15.** Plot of average R value with different (**a**) dataset sample sizes, (**b**) numbers of neurons in the hidden layer, and with (**c**) normalized or real datasets.

Figure 15b shows the average R value of test 3, which tested a different number of neurons in the hidden layer. As in test 2, test 3 used three different ANN models, which had the same input variables but different output variables. The figure shows the average R value of the three models with a different number of neurons in the hidden layer. The base case had 108 neurons in the hidden layer, which yielded an R value of 0.62. The figure shows that having fewer neurons than recommended by the literature review improved the R value. However, having fewer neurons also decreased the R value. The test shows that having 52 neurons achieved the highest R value of 0.76, which was about 1.5 times the number of input variables.

The summary of test 4 is shown in Figure 15c. This investigated the normalization of the dataset in the ANN models. As in the previous tests, three ANN models were built and tested with different configurations of the dataset. Four different datasets were tested: dataset 1 normalized both the input and output parameters, dataset 2 normalized the input but not the output, dataset 3 did not normalize the input data but normalized the output, and dataset 4 did not normalize either the input or output variables. As Figure 15c shows, normalizing both the input and output variables showed the lowest R value, and next lowest one did not normalize both the input and output. The base case (dataset 2), where the input variables were normalized and the output variable was not normalized, showed a higher R value (0.623) than the other three. However, the results demonstrated no significant difference among the different formats of the dataset. Compared to tests 2 and 3, test 4 did not show a significant improvement in the R value with a different dataset format.

Table 6 shows the overall R values among the different tests for the DF ANN model. The highest improvement in the R value of the DF ANN model was 0.856 when the number of input samples was 108, which was a 277.19% improvement from the base case (0.227). When the number of hidden neurons was 36, the R value of the DF ANN model reached the value of 0.409, which was an 81.26% improvement from the base value (0.227). The normalization of inputs and outputs had the highest R value compared with the other three normalization conditions, 9.3% more than that of the base case (0.227).

**Table 6.** Comparison between tests' R value with DF ANN model.

| TEST | | Average R Value | Ratio Compare to Base Case |
|---|---|---|---|
| No. of samples (test 2) | 20 | 0.223 | −1.66% |
| | 30 | 0.210 | −7.40% |
| | 40 | 0.234 | 3.02% |
| | 52 * | 0.227 * | 0.00% * |
| | 70 | 0.792 | 249.11% |
| | 108 | 0.856 | 277.19% |
| No. of hidden neurons (test 3) | 36 | 0.409 | 81.26% |
| | 54 | 0.354 | 56.68% |
| | 72 | 0.311 | 37.62% |
| | 90 | 0.255 | 13.09% |
| | 108 * | 0.227 * | 0.00% * |
| | 144 | 0.259 | 14.55% |
| Dataset format (test 4) | dataset 1 | 0.247 | 9.30% |
| | dataset 2 * | 0.227 * | 0.00% * |
| | dataset 3 | 0.228 | 1.07% |
| | dataset 4 | 0.247 | 9.53% |

* base case result.

As in Table 7, the order of increasing percentiles of the three tests for the QuVue ANN model had the same trend as the DF ANN model. For test 2, when the number of samples was 70, the highest R value ranked at a 52.36% improvement from the base case. For test 3, when the number of neurons was 36, it had the highest improvement of 38.65% from the base case and test 4 with the R value of 2.46%.

**Table 7.** Comparison between tests' R value with QuVue ANN model.

| TEST | | Average R Value | Ratio Compare to Base Case |
|---|---|---|---|
| No. of samples (test 2) | 20 | 0.386 | −25.40% |
| | 30 | 0.411 | −20.56% |
| | 40 | 0.449 | −13.25% |
| | 52 * | 0.518 * | 0.00% * |
| | 70 | 0.789 | 52.36% |
| | 108 | 0.765 | 47.86% |
| No. of hidden neurons (test 3) | 36 | 0.718 | 38.65% |
| | 54 | 0.659 | 27.24% |
| | 72 | 0.607 | 17.36% |
| | 90 | 0.564 | 8.91% |
| | 108 * | 0.518 * | 0.00% * |
| | 144 | 0.462 | −10.69% |
| Dataset format (test 4) | dataset 1 | 0.529 | 2.46% |
| | dataset 2 * | 0.517 * | 0.00% * |
| | dataset 3 | 0.505 | −2.20% |
| | dataset 4 | 0.525 | 1.64% |

* base case result.

Table 8 indicates the R values of three models of the WinH ANN model. As in the DF and QuVue ANN models, the R value improvement in test 2 was the most significant among the three tests, which was enhanced by 42.52% compared to the base case's R value of 0.623. The second most significant R value improvement was also seen in test 3, with an enhancement of 21.27% compared to the base case's R value of 0.623. In test 4, the R value of the base case ANN model performed best.

**Table 8.** Comparison between tests' R value with WinH ANN model.

| TEST | | Average R Value | Ratio Compare to Base Case |
|---|---|---|---|
| No. of samples (test 2) | 20 | 0.466 | −25.19% |
| | 30 | 0.500 | −19.68% |
| | 40 | 0.557 | −10.68% |
| | 52 * | 0.623 * | 0.00% * |
| | 70 | 0.854 | 37.04% |
| | 108 | 0.888 | 42.52% |
| No. of hidden neurons (test 3) | 36 | 0.729 | 16.93% |
| | 54 | 0.756 | 21.27% |
| | 72 | 0.704 | 12.99% |
| | 90 | 0.652 | 4.67% |
| | 108 * | 0.623 * | 0.00% * |
| | 144 | 0.552 | −11.35% |
| Dataset format (test 4) | dataset 1 | 0.614 | −1.38% |
| | dataset 2 * | 0.623 * | 0.00% * |
| | dataset 3 | 0.621 | −0.32% |
| | dataset 4 | 0.616 | −1.09% |

* base case result.

Figure 16 shows the best percentile of increments for each model in the three tests. Among the three tests of the percentage improvement in the R value, changing the number of samples in test 2 indicated the greatest improvement in all three ANN models (WinH, DF, and QuVue). Test 3 showed the next most efficient method to improve the R value in all three ANN models. Test 4's normalization demonstrated less efficiency compared to the other two tests, where the greatest improvement was a 9.5% improvement with the DF ANN model. However, compared to the improvement from test 2, the normalization impact on the improvement was limited.

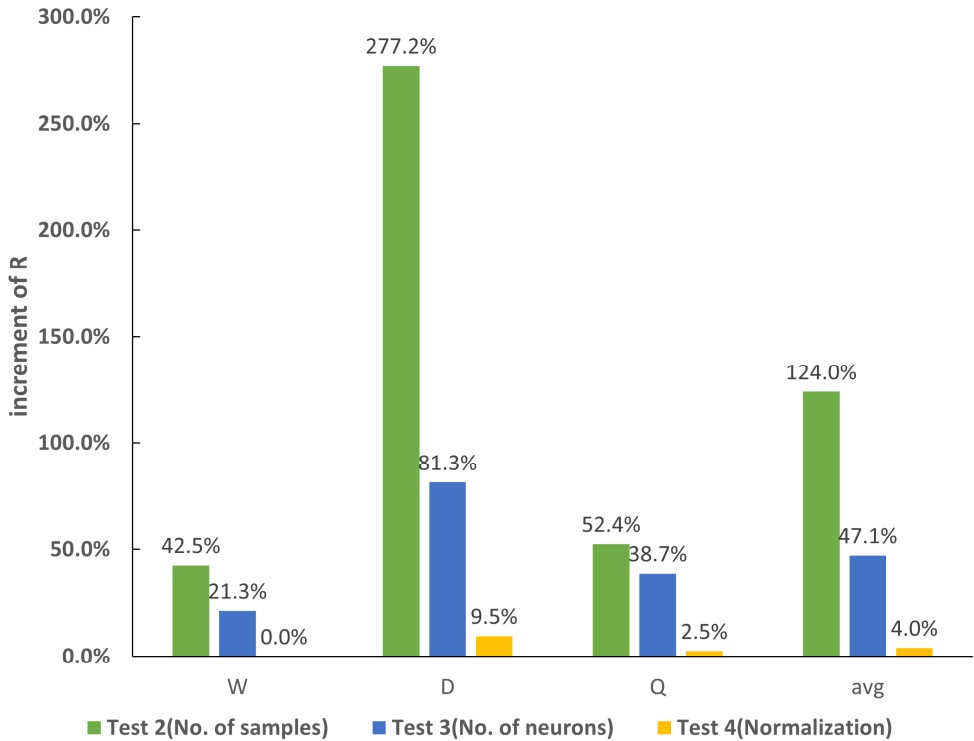

**Figure 16.** The percentile of increment for each model in three tests.

## 4. Discussion

Employing the same setups, including the training function, 52 samples, 108 hidden neurons, and 36 input parameters, test 1 found that the integration of the five performance metric prediction models (model C) had the highest accuracy performance value. The single-objective prediction model (model A) was a less recommended model because the average R value was the lowest. Combining all of the output parameters together not only improved the accuracy performance of building an ANN model, but also reduced the amount of work compared to building many models. The obtained results were not completely consistent with our initial expectations. One possible explanation for the lower performance of a single output compared to multiple outputs could be that the outcomes are more closely related to each other, which in turn may impact the final result. We see this as an area for further investigation. Given that our team's expertise does not lie in the theory of ANNs, we plan to collaborate with mathematicians or computer science experts to explore this question further. There is few research on this topic.

The quantity of samples was tested in training the ANN models in test 2. In the DF ANN model, when the number of samples increased from 52 to 108, the R value improved by 277.19%, which was the highest percentile improvement in the three models of test 2. The improvement in the R value rose from 0.5175 to 0.7885, almost 52.4%, in the QuVue ANN model, ranking second. The percentile increment of the WinH ANN was 42.5% with an R value from 0.623 to 0.888. When the number of samples (72 for QuVue and 108 for both WinH and DF) was two or three times the number of variables, the accuracy of the ANN models performed best. In Figure 15a, when the number of samples was 72, the R value was 0.85. However, increasing the number of samples will not dramatically improve the average R value. By balancing the model's complexity and training time, two times the number of input variables may be recommended. In previous research, Rocío Escandón et al. concluded that the sample size ensured reliable prediction results according to a specific building category, which means that a ratio of 2.5 between the sample size and the number of characteristic parameters (input variables) can increase the accuracy [24].

Our result showed that 2.5 may be a reasonable conclusion, but based on our test, in some cases 2.0 also can be used.

Based on test 3, the correlation coefficient (R) of the DF was improved from 0.227 to 0.409 (81.3%), thus ranking first. The average R value of QuVue improved to 0.718, 38.7% more than that of the base case, and the improvement in WinH ranked lowest. Moreover, the highest value of R was attained when the number of hidden neurons of 36 for DF and QuVue or 54 for WinH were selected. As can be seen in Figure 15b, there existed a peak value in the average R value of the three models. Based on the test, when the number of hidden neurons was about 1.5 times the number of input variables, the ANN models had a better accuracy performance than the other models. Many researchers investigated the same topic in previous studies. Sofuoglu tested eight feedforward networks with different hidden layers and numbers of neurons to search for better performance models according to R, $R^2$, and RMSE to predict the prevalence of building-related symptoms (BRSs) of office building occupants [46]. Ashtiani et al., using a varied number of hidden neurons from 1 to 100, showed that 10 hidden neurons achieved the best network performance [37]. Kerdan and Gálvez demonstrated that the number of hidden neurons is insensitive to the performance of the ANN model [47]. Moon and Jung took three steps to find the optimal artificial neural network model in terms of $R^2$ to predict the setback temperature of a building [48]. The optimal values for the number of hidden layers, number of hidden neurons, learning rate, and moment were found to be 4, 9, 0.6, and 0.9, respectively. Wang et al. discussed the effect of the time intervals of inputs and the number of hidden neurons on model accuracy using the global sensitivity analysis method for air heat pump operation systems [49]. Only a limited number of studies have investigated the link between the number of hidden neurons and the number of input variables, but test 3 did confirm a pattern concerning the accuracy of ANN model predictions.

The test of the normalization combination of variables (test 4) demonstrated little influence on the improvement in accuracy of the three models. Furthermore, the highest percentile increments were derived from the different combinations of the input and output variables in different models. From the test result, it was difficult to generalize a certain trend; the highest percentile improvement was 9.53% when the original inputs and outputs of the variables were selected in the DF ANN. In the QuVue ANN, both the normalized input and output parameters showed the highest improvement in the R value of 2.5%. Additionally, the base case (normalization input with the original variables) in the WinH ANN had a higher R value than the other three conditions. Therefore, the normalization of the input and output variables was insensitive to the accuracy performance of the ANN models [50]. In previous studies, the normalized method was commonly conducted to construct the ANN models; however, the normalization combination of the input and output variables was seldom investigated regarding the accuracy performance of the models.

From Figure 16, we can observe the summary of the percentile improvements in the three tests, which were the number of input samples, the number of hidden neurons, and the normalization combination of the variables. Among the three tests, changing the number of samples in the ANN models yielded the highest R value compared to base case.

## 5. Conclusions

This study investigated a strategy to construct a more reliable ANN model to predict a building's indoor and outdoor performance measures. A sensitivity analysis was utilized to test different strategies to build the ANN model, namely the number of output variables, the number of samples, the number of hidden neurons, and the normalization combination of the variables, that would impact the prediction of the accuracy performance. The correlation coefficients (R) of each ANN model were examined. A summary of the findings is as follows:

(1) The ideal method to build an ANN model that has the same input variables was to see if combining the performance metrics as the output variables demonstrated better prediction accuracy than modeling the ANN separately with each output variable. However, the performance indices depended on the statistical properties of the data due to the research limitations.

(2) The number of samples of input variables was sensitive to the accuracy performance of the ANN models. This relationship between the number of input variables and the number of training datasets was not linear. There existed a point where the linear curve line plateaued. The study found that two times the number of input variables in the quantity of training datasets can lead to a high accuracy of prediction.

(3) Increasing the number of hidden neurons usually led to the decreasing accuracy performance of the ANN models. However, too many hidden neurons did not further improve accuracy and even reduced it. The ideal number of neurons in the hidden layer was approximately 1.5 times the number of input variables based on the training models of R.

(4) The normalization of the input and output variables did not show a significant improvement in accuracy from the test.

(5) From Figure 16, test 2 showed the best R value. Therefore, it is possible to give an order of priority in building an ANN model. Firstly, it is possible to increase the number of dataset samples. Secondly, it is advised to increase the number of hidden neurons, and normalization is the last step to improving accuracy.

It is important to address the limitations of this work and that a careful interpretation of the findings be utilized. This study tried to capture various scenarios in building environmental conditions for generalized application. However, we acknowledge the limited scenarios of building models that were tested. We also understand the problem of overfitting in establishing ANN models. This paper did not fully investigate this matter and requires work with ridge regression and LASSO (least absolute shrinkage and selection operator) to investigate further. Previous research has indicated that decreasing the number of neurons below the input quantity can enhance the prediction accuracy of artificial neural network (ANN) models, as evidenced by higher R values. While the findings in Tables 2 and 3 of this paper support this trend, demonstrating the highest accuracy at 36 neurons, this suggests that increasing the number of neurons may diminish the model's predictive capability. To gain a more comprehensive understanding of this trend, future experiments should consider a broader range of neuron quantities for analysis. In this study, the variation of the number of hidden neurons in the ANN was explored within a range of 1–3 times the independent variable parameters. For investigations extending beyond this multiplicative relationship, further discussions and studies are warranted.

For every test, although the training procedure involved conducting 30 trials for all ANN models, the average value was used to compare among the different types of datasets to consider result validation. We will proceed with further analysis using cross-validation, specifically employing k-fold methods.

This study's tests only considered specific elements of the ANN models and further work is necessary to include other factors of constructing ANN models, such as the number of hidden layers, training algorithms, transferring algorithms, learning rate, and the moment, to enhance the performance of ANN models. The geometry variables were only considered as the input variables for this test, and further research is needed to include other input variables such as material properties. Also, the tests were based on one test case, and it would be better to include different test cases to generalize the findings.

**Author Contributions:** Conceptualization, S.W. and Y.K.Y.; methodology, S.W. and Y.K.Y.; software, S.W. and Y.K.Y.; validation, Y.K.Y.; formal analysis, S.W.; investigation, Y.K.Y.; resources, N.L.; data curation, S.W. and Y.K.Y.; writing—original draft preparation, S.W.; writing—review and editing, Y.K.Y.; visualization, S.W.; supervision, Y.K.Y. and N.L.; project administration, Y.K.Y. and N.L.; funding acquisition, N.L. All authors have read and agreed to the published version of the manuscript.

**Funding:** This research was funded by National Key R&D Program of China (2022YFC3803801 in 2022YFC3803800); Beijing Postdoctoral Research Foundation (2023-zz-143); Beijing Key Laboratory of Green Building and Energy-Efficiency Technology Open Foundation.

**Data Availability Statement:** https://pan.baidu.com/s/19vEQ6uog44ekCwu425FdFQ?pwd=1t32, accessed on 28 October 2023.

**Conflicts of Interest:** The authors declare no conflict of interest.

## Appendix A

**Table A1.** The correlation coefficients of the predicted results of D_NN for different numbers of samples.

| Trials | 20 | 30 | 40 | 52 | 70 | 108 |
|--------|------|------|------|------|------|------|
| 1 | 0.0188 | 0.1855 | 0.1972 | 0.1470 | 0.8788 | 0.9026 |
| 2 | 0.1395 | 0.3126 | 0.3501 | 0.1614 | 0.8060 | 0.8550 |
| 3 | 0.3096 | 0.2885 | 0.2271 | 0.2711 | 0.6647 | 0.8821 |
| 4 | 0.0504 | 0.2291 | 0.3155 | 0.2328 | 0.7604 | 0.8856 |
| 5 | 0.2988 | 0.1632 | 0.2287 | 0.2238 | 0.7900 | 0.8736 |
| 6 | 0.1329 | 0.2202 | 0.1982 | 0.2461 | 0.8295 | 0.8544 |
| 7 | 0.2111 | 0.2573 | 0.2573 | 0.1762 | 0.7708 | 0.8631 |
| 8 | 0.0899 | 0.2218 | 0.3230 | 0.1848 | 0.8042 | 0.8236 |
| 9 | 0.3336 | 0.0183 | 0.1749 | 0.3535 | 0.8357 | 0.7970 |
| 10 | 0.1870 | 0.2392 | 0.2530 | 0.3126 | 0.8932 | 0.8800 |
| 11 | 0.3035 | 0.2583 | 0.2398 | 0.1075 | 0.7762 | 0.8373 |
| 12 | 0.2975 | 0.1596 | 0.2800 | 0.2128 | 0.7623 | 0.8351 |
| 13 | 0.0466 | 0.2958 | 0.1407 | 0.3411 | 0.8230 | 0.9303 |
| 14 | 0.3366 | 0.1636 | 0.2354 | 0.1093 | 0.8211 | 0.8775 |
| 15 | 0.2978 | 0.1069 | 0.2487 | 0.3191 | 0.7416 | 0.9318 |
| 16 | 0.2223 | 0.1867 | 0.2056 | 0.2129 | 0.8206 | 0.8824 |
| 17 | 0.3950 | 0.1952 | 0.3609 | 0.2893 | 0.7800 | 0.9153 |
| 18 | 0.3583 | 0.3000 | 0.2720 | 0.3134 | 0.6747 | 0.8164 |
| 19 | 0.4499 | 0.2178 | 0.1180 | 0.1595 | 0.9013 | 0.9083 |
| 20 | 0.2205 | 0.1789 | 0.2236 | 0.2884 | 0.6889 | 0.8185 |
| 21 | 0.1894 | 0.2580 | 0.2945 | 0.1288 | 0.7355 | 0.7946 |
| 22 | 0.2422 | 0.1877 | 0.3235 | 0.2373 | 0.7078 | 0.8788 |
| 23 | 0.1129 | 0.1660 | 0.1250 | 0.2627 | 0.7393 | 0.9054 |
| 24 | 0.2602 | 0.1059 | 0.2336 | 0.2262 | 0.8353 | 0.6396 |
| 25 | 0.2425 | 0.3335 | 0.2706 | 0.1976 | 0.8357 | 0.8043 |
| 26 | 0.4117 | 0.1720 | 0.2597 | 0.3191 | 0.8560 | 0.8957 |
| 27 | 0.2458 | 0.2519 | 0.1160 | 0.1396 | 0.7926 | 0.8464 |
| 28 | 0.1131 | 0.2542 | 0.1359 | 0.2765 | 0.7989 | 0.8921 |
| 29 | 0.2161 | 0.1640 | 0.1690 | 0.1284 | 0.8427 | 0.7878 |
| 30 | 0.2138 | 0.3077 | 0.1737 | 0.1947 | 0.7900 | 0.9262 |
| Avg | 0.2231 | 0.2101 | 0.2337 | 0.2269 | 0.7920 | 0.8557 |

**Table A2.** The correlation coefficients of the predicted results of Q_NN for different numbers of samples.

| Trials | 20 | 30 | 40 | 52 | 70 | 108 |
|--------|------|------|------|------|------|------|
| 1 | 0.4219 | 0.3951 | 0.4614 | 0.5178 | 0.8168 | 0.7242 |
| 2 | 0.4056 | 0.3727 | 0.4309 | 0.5081 | 0.7738 | 0.8423 |
| 3 | 0.3731 | 0.3685 | 0.3934 | 0.5177 | 0.8037 | 0.7921 |
| 4 | 0.4845 | 0.5412 | 0.4426 | 0.5327 | 0.7843 | 0.7808 |
| 5 | 0.3736 | 0.4413 | 0.4347 | 0.5751 | 0.7794 | 0.7690 |
| 6 | 0.4378 | 0.4723 | 0.4369 | 0.5342 | 0.8806 | 0.7667 |
| 7 | 0.3951 | 0.3269 | 0.4419 | 0.4949 | 0.8055 | 0.8032 |

**Table A2.** *Cont.*

| Trials | 20 | 30 | 40 | 52 | 70 | 108 |
|--------|------|------|------|------|------|------|
| 8 | 0.4363 | 0.5038 | 0.4639 | 0.4915 | 0.6429 | 0.8173 |
| 9 | 0.4178 | 0.3715 | 0.4672 | 0.5124 | 0.7195 | 0.7900 |
| 10 | 0.5499 | 0.3883 | 0.4843 | 0.5571 | 0.7774 | 0.5913 |
| 11 | 0.3938 | 0.4564 | 0.4720 | 0.3734 | 0.8290 | 0.7203 |
| 12 | 0.5078 | 0.4399 | 0.4112 | 0.4914 | 0.6024 | 0.8688 |
| 13 | 0.2033 | 0.5138 | 0.4205 | 0.5279 | 0.8485 | 0.8387 |
| 14 | 0.4194 | 0.4746 | 0.3403 | 0.4819 | 0.8588 | 0.7497 |
| 15 | 0.3526 | 0.4044 | 0.5141 | 0.5254 | 0.8030 | 0.6779 |
| 16 | 0.4173 | 0.4742 | 0.4832 | 0.5841 | 0.7287 | 0.7045 |
| 17 | 0.0435 | 0.3582 | 0.3802 | 0.4996 | 0.8337 | 0.8388 |
| 18 | 0.4505 | 0.4089 | 0.5396 | 0.4942 | 0.9035 | 0.8438 |
| 19 | 0.4690 | 0.3479 | 0.4590 | 0.4898 | 0.6988 | 0.7519 |
| 20 | 0.2469 | 0.4397 | 0.3885 | 0.5797 | 0.8411 | 0.6340 |
| 21 | 0.3673 | 0.4159 | 0.4245 | 0.5355 | 0.8180 | 0.8026 |
| 22 | 0.3026 | 0.3706 | 0.4517 | 0.5610 | 0.7827 | 0.7578 |
| 23 | 0.3901 | 0.4811 | 0.4718 | 0.5297 | 0.7716 | 0.8409 |
| 24 | 0.3645 | 0.3303 | 0.4143 | 0.5384 | 0.8054 | 0.7319 |
| 25 | 0.3618 | 0.4387 | 0.5004 | 0.4865 | 0.7997 | 0.8194 |
| 26 | 0.4432 | 0.3225 | 0.5099 | 0.5354 | 0.8465 | 0.5977 |
| 27 | 0.4954 | 0.3786 | 0.4901 | 0.4215 | 0.7762 | 0.8180 |
| 28 | 0.3415 | 0.4011 | 0.4600 | 0.5000 | 0.7455 | 0.7523 |
| 29 | 0.3862 | 0.3456 | 0.4900 | 0.5831 | 0.6845 | 0.5786 |
| 30 | 0.3300 | 0.3500 | 0.3900 | 0.5452 | 0.8554 | 0.7891 |
| Avg | 0.3861 | 0.4111 | 0.4489 | 0.5175 | 0.7885 | 0.7652 |

**Table A3.** The correlation coefficients of the predicted results of W_NN for different numbers of samples.

| Trials | 20 | 30 | 40 | 52 | 70 | 108 |
|--------|------|------|------|------|------|------|
| 1 | 0.4755 | 0.5120 | 0.4793 | 0.6293 | 0.9064 | 0.8744 |
| 2 | 0.4890 | 0.4836 | 0.5928 | 0.5755 | 0.8433 | 0.8789 |
| 3 | 0.4566 | 0.5513 | 0.5163 | 0.6167 | 0.8734 | 0.8440 |
| 4 | 0.4350 | 0.5416 | 0.5844 | 0.6298 | 0.8571 | 0.8900 |
| 5 | 0.5001 | 0.4687 | 0.5214 | 0.6430 | 0.9039 | 0.9134 |
| 6 | 0.3520 | 0.4925 | 0.5805 | 0.6461 | 0.8389 | 0.8991 |
| 7 | 0.4722 | 0.4404 | 0.6001 | 0.5954 | 0.8208 | 0.8754 |
| 8 | 0.4037 | 0.5347 | 0.4707 | 0.6292 | 0.8486 | 0.8875 |
| 9 | 0.4323 | 0.5003 | 0.5723 | 0.5568 | 0.8548 | 0.9149 |
| 10 | 0.4305 | 0.4690 | 0.5692 | 0.6548 | 0.8606 | 0.9181 |
| 11 | 0.4429 | 0.4893 | 0.6102 | 0.6149 | 0.8939 | 0.8947 |
| 12 | 0.5389 | 0.5064 | 0.5628 | 0.6283 | 0.8002 | 0.8751 |
| 13 | 0.4161 | 0.4635 | 0.6139 | 0.6104 | 0.8003 | 0.9000 |
| 14 | 0.5540 | 0.5458 | 0.5070 | 0.6341 | 0.8603 | 0.8965 |
| 15 | 0.5140 | 0.5130 | 0.5859 | 0.6211 | 0.8495 | 0.8612 |
| 16 | 0.4570 | 0.4443 | 0.5870 | 0.6022 | 0.8827 | 0.8997 |
| 17 | 0.5665 | 0.5193 | 0.4885 | 0.6420 | 0.8361 | 0.8898 |
| 18 | 0.4908 | 0.4647 | 0.5343 | 0.6350 | 0.8673 | 0.9154 |
| 19 | 0.3855 | 0.4356 | 0.5163 | 0.6171 | 0.8908 | 0.8814 |
| 20 | 0.4789 | 0.5009 | 0.5561 | 0.6545 | 0.8526 | 0.8728 |
| 21 | 0.5081 | 0.5160 | 0.5910 | 0.6492 | 0.8298 | 0.8766 |
| 22 | 0.4559 | 0.5785 | 0.5638 | 0.6192 | 0.8032 | 0.9085 |
| 23 | 0.4142 | 0.5241 | 0.5667 | 0.5926 | 0.8325 | 0.9317 |
| 24 | 0.4533 | 0.4956 | 0.5829 | 0.6346 | 0.8715 | 0.8853 |
| 25 | 0.4782 | 0.5704 | 0.5633 | 0.5997 | 0.8227 | 0.8539 |
| 26 | 0.5211 | 0.5348 | 0.5790 | 0.5974 | 0.8372 | 0.8725 |
| 27 | 0.5110 | 0.4500 | 0.5413 | 0.6024 | 0.8593 | 0.8666 |

**Table A3.** *Cont.*

| Trials | 20 | 30 | 40 | 52 | 70 | 108 |
|--------|--------|--------|--------|--------|--------|--------|
| 28 | 0.3814 | 0.4858 | 0.5462 | 0.6436 | 0.9181 | 0.8680 |
| 29 | 0.5231 | 0.4711 | 0.5579 | 0.6793 | 0.8460 | 0.9044 |
| 30 | 0.4467 | 0.5096 | 0.5542 | 0.6378 | 0.8674 | 0.8902 |
| Avg | 0.4661 | 0.5004 | 0.5565 | 0.6231 | 0.8539 | 0.8880 |

**Table A4.** The correlation coefficients (R) of the predicted results of D_NN for different numbers of hidden neurons.

| Trials | 36 | 54 | 72 | 90 | 108 | 144 |
|--------|--------|--------|--------|--------|--------|--------|
| 1 | 0.4660 | 0.3207 | 0.3681 | 0.2689 | 0.1470 | 0.1692 |
| 2 | 0.4153 | 0.4513 | 0.3673 | 0.1800 | 0.1614 | 0.2436 |
| 3 | 0.4182 | 0.3186 | 0.3221 | 0.2213 | 0.2711 | 0.1005 |
| 4 | 0.4389 | 0.3856 | 0.3353 | 0.2470 | 0.2328 | 0.2664 |
| 5 | 0.4064 | 0.1944 | 0.2849 | 0.2488 | 0.2238 | 0.3255 |
| 6 | 0.4251 | 0.3873 | 0.3623 | 0.3336 | 0.2461 | 0.2925 |
| 7 | 0.4029 | 0.3085 | 0.3644 | 0.3187 | 0.1762 | 0.3704 |
| 8 | 0.4969 | 0.3823 | 0.3225 | 0.2691 | 0.1848 | 0.2535 |
| 9 | 0.2224 | 0.2535 | 0.3677 | 0.2403 | 0.3535 | 0.2692 |
| 10 | 0.4076 | 0.2655 | 0.3574 | 0.2614 | 0.3126 | 0.2716 |
| 11 | 0.3072 | 0.3824 | 0.3291 | 0.1806 | 0.1075 | 0.2995 |
| 12 | 0.4436 | 0.5089 | 0.1687 | 0.1571 | 0.2128 | 0.3308 |
| 13 | 0.4030 | 0.2725 | 0.2577 | 0.2349 | 0.3411 | 0.3387 |
| 14 | 0.4745 | 0.3053 | 0.3325 | 0.2423 | 0.1093 | 0.2002 |
| 15 | 0.4322 | 0.4010 | 0.2322 | 0.2710 | 0.3191 | 0.2089 |
| 16 | 0.3562 | 0.4450 | 0.3426 | 0.3727 | 0.2129 | 0.2270 |
| 17 | 0.3780 | 0.3800 | 0.3757 | 0.0901 | 0.2893 | 0.2866 |
| 18 | 0.4266 | 0.3930 | 0.2298 | 0.3055 | 0.3134 | 0.2121 |
| 19 | 0.4716 | 0.3427 | 0.2259 | 0.2607 | 0.1595 | 0.3176 |
| 20 | 0.4444 | 0.4076 | 0.2944 | 0.2604 | 0.2884 | 0.2253 |
| 21 | 0.3587 | 0.4088 | 0.3346 | 0.3300 | 0.1288 | 0.3124 |
| 22 | 0.4450 | 0.3225 | 0.3080 | 0.2913 | 0.2373 | 0.2175 |
| 23 | 0.3015 | 0.4189 | 0.3436 | 0.2960 | 0.2627 | 0.2246 |
| 24 | 0.5069 | 0.3557 | 0.2334 | 0.2213 | 0.2262 | 0.2381 |
| 25 | 0.4425 | 0.2728 | 0.2705 | 0.2758 | 0.1976 | 0.2670 |
| 26 | 0.4732 | 0.4288 | 0.3502 | 0.2399 | 0.3191 | 0.1298 |
| 27 | 0.3735 | 0.2223 | 0.3195 | 0.2430 | 0.1396 | 0.2456 |
| 28 | 0.3005 | 0.3079 | 0.3001 | 0.2677 | 0.2765 | 0.3566 |
| 29 | 0.4296 | 0.4152 | 0.2944 | 0.2706 | 0.1284 | 0.2991 |
| 30 | 0.4329 | 0.3685 | 0.8698 | 0.2605 | 0.1947 | 0.2591 |
| Avg | 0.4093 | 0.3538 | 0.3107 | 0.2553 | 0.2258 | 0.2586 |

**Table A5.** The correlation coefficients (R) of the predicted results of Q_NN for different numbers of hidden neurons.

| Trials | 36 | 54 | 72 | 90 | 108 | 144 |
|--------|--------|--------|--------|--------|--------|--------|
| 1 | 0.7527 | 0.7276 | 0.5908 | 0.5733 | 0.5178 | 0.4628 |
| 2 | 0.7024 | 0.5618 | 0.6045 | 0.6409 | 0.5081 | 0.4541 |
| 3 | 0.6667 | 0.6455 | 0.5968 | 0.5973 | 0.5177 | 0.4511 |
| 4 | 0.6254 | 0.6695 | 0.5865 | 0.4684 | 0.5327 | 0.3933 |
| 5 | 0.7287 | 0.6849 | 0.6308 | 0.5621 | 0.5751 | 0.5361 |
| 6 | 0.6190 | 0.6235 | 0.6076 | 0.5174 | 0.5342 | 0.4677 |
| 7 | 0.7408 | 0.6642 | 0.6103 | 0.5971 | 0.4949 | 0.5404 |
| 8 | 0.7323 | 0.5689 | 0.5734 | 0.6133 | 0.4915 | 0.5287 |
| 9 | 0.7964 | 0.6905 | 0.6201 | 0.5803 | 0.5124 | 0.4713 |
| 10 | 0.6083 | 0.6884 | 0.6275 | 0.5982 | 0.5571 | 0.3499 |

**Table A5.** *Cont.*

| Trials | 36 | 54 | 72 | 90 | 108 | 144 |
|--------|------|------|------|------|------|------|
| 11 | 0.7103 | 0.6760 | 0.5564 | 0.5144 | 0.3734 | 0.3986 |
| 12 | 0.6586 | 0.6526 | 0.5500 | 0.5645 | 0.4914 | 0.5046 |
| 13 | 0.7169 | 0.6835 | 0.5529 | 0.5500 | 0.5279 | 0.4789 |
| 14 | 0.7460 | 0.7313 | 0.5963 | 0.5304 | 0.4819 | 0.4776 |
| 15 | 0.7451 | 0.6831 | 0.6464 | 0.5729 | 0.5254 | 0.4924 |
| 16 | 0.7484 | 0.6410 | 0.6741 | 0.5998 | 0.5841 | 0.4482 |
| 17 | 0.7434 | 0.6258 | 0.6300 | 0.4867 | 0.4996 | 0.4654 |
| 18 | 0.7255 | 0.6384 | 0.5047 | 0.5658 | 0.4942 | 0.4902 |
| 19 | 0.6989 | 0.6312 | 0.5353 | 0.5461 | 0.4898 | 0.5080 |
| 20 | 0.7342 | 0.6478 | 0.6538 | 0.5276 | 0.5797 | 0.4931 |
| 21 | 0.7208 | 0.6063 | 0.6129 | 0.5375 | 0.5355 | 0.3973 |
| 22 | 0.7558 | 0.7317 | 0.6072 | 0.5653 | 0.5610 | 0.3710 |
| 23 | 0.7324 | 0.6627 | 0.6248 | 0.5762 | 0.5297 | 0.4733 |
| 24 | 0.7990 | 0.6443 | 0.5788 | 0.5696 | 0.5384 | 0.5477 |
| 25 | 0.6501 | 0.6232 | 0.6539 | 0.5399 | 0.4865 | 0.3436 |
| 26 | 0.740 8 | 0.7129 | 0.6839 | 0.6473 | 0.5354 | 0.5031 |
| 27 | 0.7854 | 0.6125 | 0.6159 | 0.5757 | 0.4215 | 0.4274 |
| 28 | 0.7328 | 0.6472 | 0.5886 | 0.4934 | 0.5000 | 0.4736 |
| 29 | 0.6917 | 0.7033 | 0.6990 | 0.5355 | 0.5831 | 0.4549 |
| 30 | 0.7156 | 0.6755 | 0.5887 | 0.5834 | 0.5452 | 0.4803 |
| Avg | 0.7175 | 0.6585 | 0.6073 | 0.5636 | 0.5175 | 0.4622 |

**Table A6.** The correlation coefficients of the predicted results of W_NN for different numbers of hidden neurons.

| Trials | 36 | 54 | 72 | 90 | 108 | 144 |
|--------|------|------|------|------|------|------|
| 1 | 0.7670 | 0.7690 | 0.7290 | 0.6327 | 0.6293 | 0.5265 |
| 2 | 0.7017 | 0.7633 | 0.7353 | 0.6569 | 0.5755 | 0.5470 |
| 3 | 0.7274 | 0.7245 | 0.6967 | 0.6320 | 0.6167 | 0.5986 |
| 4 | 0.7358 | 0.7513 | 0.6593 | 0.6382 | 0.6298 | 0.5421 |
| 5 | 0.8041 | 0.7478 | 0.6855 | 0.5963 | 0.6430 | 0.5871 |
| 6 | 0.7565 | 0.7530 | 0.7260 | 0.6642 | 0.6461 | 0.5780 |
| 7 | 0.7407 | 0.7534 | 0.6795 | 0.6589 | 0.5954 | 0.5613 |
| 8 | 0.7808 | 0.7364 | 0.6706 | 0.6928 | 0.6292 | 0.5489 |
| 9 | 0.7932 | 0.7825 | 0.7287 | 0.6210 | 0.5568 | 0.5149 |
| 10 | 0.6418 | 0.7661 | 0.7202 | 0.6928 | 0.6548 | 0.5944 |
| 11 | 0.7736 | 0.7751 | 0.7083 | 0.6409 | 0.6149 | 0.5195 |
| 12 | 0.7686 | 0.7710 | 0.7131 | 0.6457 | 0.6283 | 0.5926 |
| 13 | 0.7227 | 0.7836 | 0.7186 | 0.6887 | 0.6104 | 0.5148 |
| 14 | 0.7536 | 0.7759 | 0.7089 | 0.6588 | 0.6341 | 0.5176 |
| 15 | 0.6542 | 0.7107 | 0.6705 | 0.6977 | 0.6211 | 0.5131 |
| 16 | 0.7913 | 0.7403 | 0.6842 | 0.6618 | 0.6022 | 0.5546 |
| 17 | 0.7840 | 0.7566 | 0.6919 | 0.6321 | 0.6420 | 0.5773 |
| 18 | 0.6381 | 0.7504 | 0.6900 | 0.6905 | 0.6350 | 0.5062 |
| 19 | 0.6574 | 0.7643 | 0.7025 | 0.6079 | 0.6171 | 0.5461 |
| 20 | 0.6402 | 0.7659 | 0.7342 | 0.6882 | 0.6545 | 0.5202 |
| 21 | 0.7780 | 0.7694 | 0.6938 | 0.5584 | 0.6492 | 0.5667 |
| 22 | 0.7887 | 0.7773 | 0.6782 | 0.6764 | 0.6192 | 0.6235 |
| 23 | 0.6744 | 0.7609 | 0.7252 | 0.6789 | 0.5926 | 0.5231 |
| 24 | 0.7875 | 0.6325 | 0.7342 | 0.6364 | 0.6346 | 0.5537 |
| 25 | 0.6422 | 0.7772 | 0.6976 | 0.6395 | 0.5997 | 0.5554 |
| 26 | 0.6304 | 0.7685 | 0.6859 | 0.6600 | 0.5974 | 0.5402 |
| 27 | 0.7283 | 0.7476 | 0.7528 | 0.6274 | 0.6024 | 0.5522 |
| 28 | 0.7586 | 0.7565 | 0.7328 | 0.6619 | 0.6436 | 0.5210 |
| 29 | 0.6799 | 0.7742 | 0.6937 | 0.6575 | 0.6793 | 0.6023 |
| 30 | 0.7564 | 0.7633 | 0.6725 | 0.6705 | 0.6378 | 0.5723 |
| Avg | 0.7286 | 0.7556 | 0.7040 | 0.6522 | 0.6231 | 0.5524 |

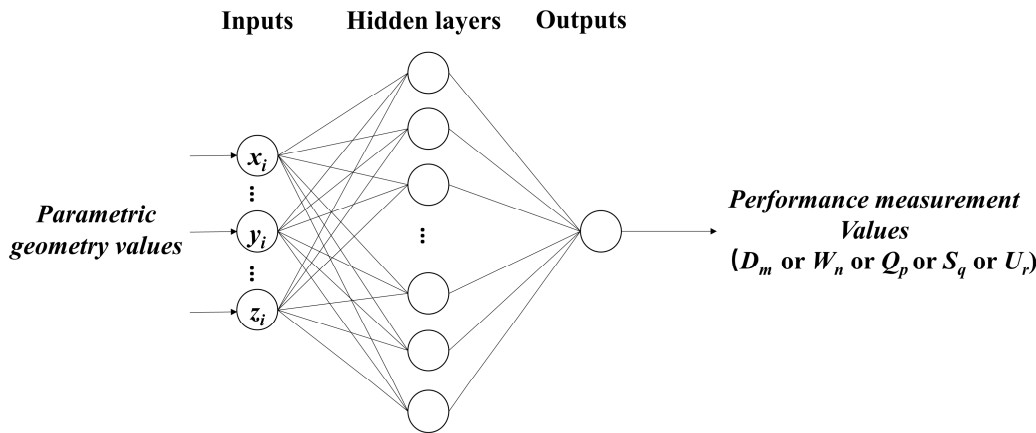

**Figure A1.** Model A: ANN model structure.

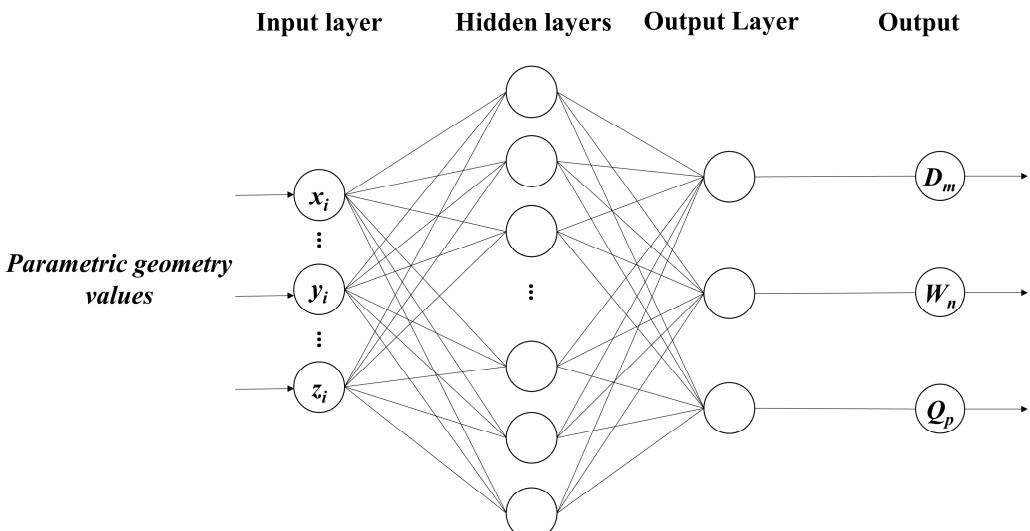

**Figure A2.** Model B group 1: ANN model for indoor measurements.

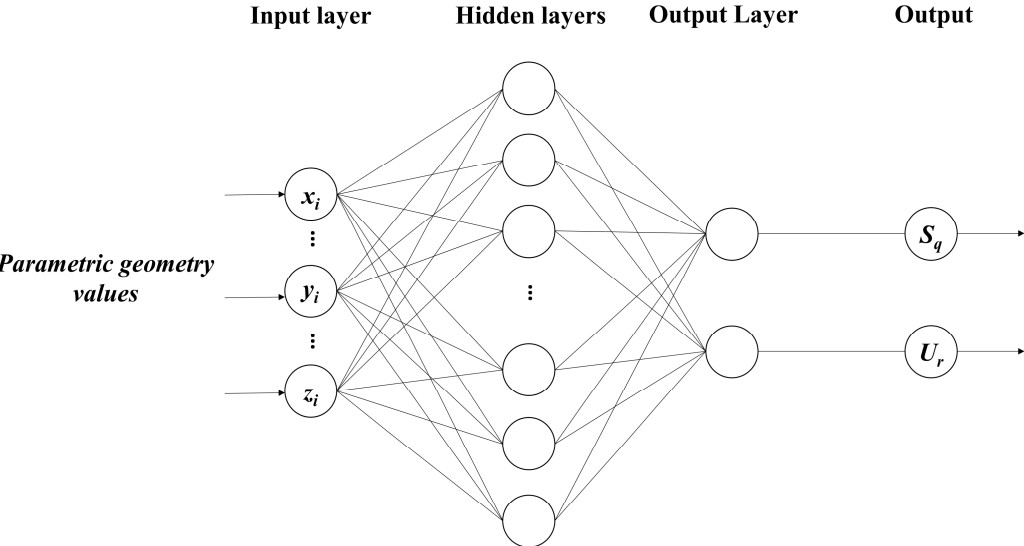

**Figure A3.** Model B group 2: ANN model with 2 outdoor measurements.

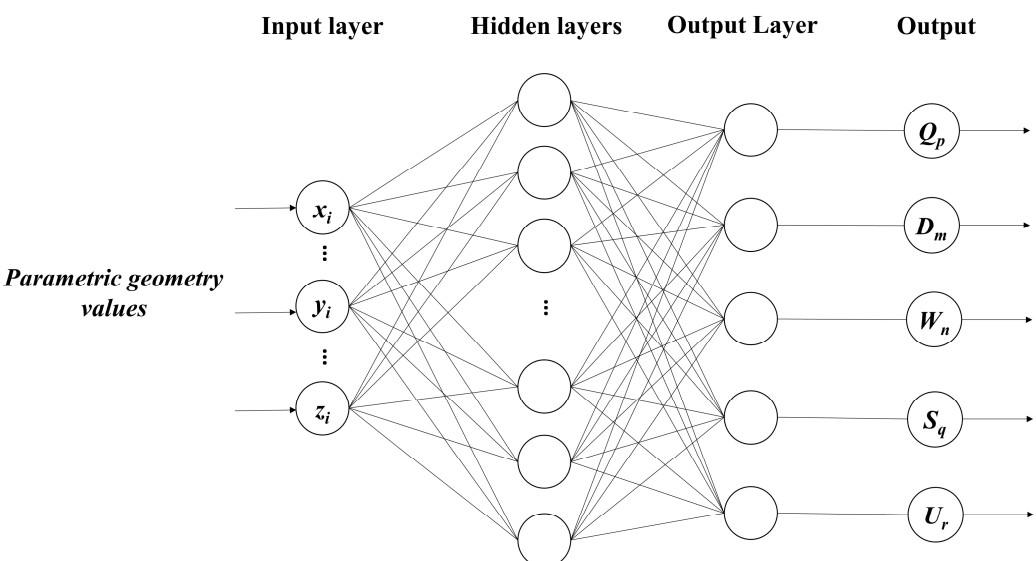

**Figure A4.** Model C: ANN model structure.

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
