# Peer review of "The ANN Architecture Analysis: A Case Study on Daylight, Visual, and Outdoor Thermal Metrics of Residential Buildings in China"

_buildings, doi:10.3390/buildings13112795_

Round 1

Reviewer 1 Report

Comments and Suggestions for Authors

The paper submitted for review, entitled Artificial Neural Network (ANN) Model's Sensitivity Analysis: Focus on Daylight Factor (DF), Sunlight hours, QuVue (Sky View Ratio) and Universal Thermal Climate Index (UTCI) of residential buildings, a case study in China, addresses a very important and timely issue. The indicators analysed in the study can be helpful in determining the energy needs of buildings. The topics of the study are within the scope of the journal Buildings.
Despite its many merits, I believe that the work needs to be revised.
1. The title of the study is long and too detailed.
2. Introduction: The literature review is short. It does not provide sufficient justification to undertake further research in this area. The literature items on increasing ANN prediction performance are mainly from 2009-2015. One item from 2018 and one from 2019. There are no items from the last three years in the entire literature.
What guided the selection of indicators for the analysis? Daylight Factor (DF) Sky view ratio (QuVue) Window Sunlight Hours(WinH) Site Sunlight Hours(SiteH) Universal Thermal Climate Index (UTCI)
How was the potential list of independent variables determined?
No precise indication of what will be new to this work.
3. Materials and Methods
Figure 2 should be preceded by an introductory text on the methodology of the survey. The diagram itself is good but lacks description. Maybe show the diagram as a summary of the survey implementation methodology?
Table 1. separate Input Variables and Output Variables by a line.
What are the assumptions made in the work? The buildings were controlled not to intersect with other buildings and buildings were not to move beyond the site boundary, and the total building floor area was between ±10% of the original case. The test buildings were able to move in both x and y axes and the possible moveable range from -10 to 20m was based on the initial building location. The average height of a floor was 3.1m. The building height (z) can change between the heights from 34.1 to 99.2m.
Figure 4. Why are there no variables between - 20 and - 15?
Why are simulation results presented in the methodology chapter? (Figure 4 and Figure 7).
What could the input variables have been? Their number greater than 40.
4. Results
Assess the impact of normalisation. Please check whether normalisation is not performed automatically in the Matlab algorithm for ANN. If so, this part of the study is devoid of purpose.
Can the results obtained be generalised? Do they characterise only this study? Complete lack of information on the variability of the model input features. No information on how and why the independent variables were selected for the different indicators (DF, QuVue, WinH, SiteH, UTCI).
I believe that it is also advisable to show quality assessment indicators other than R2. With similar results, they will allow a better solution to be identified.
What is missing from the paper is an assessment of the significance of the differences between the results obtained. The information that something is bigger or smaller is not sufficient. We need to show whether there is or is not a difference.
Line 555. How do you explain that for an analogous set of independent variables a model with one output is less accurate than one with several?
In conclusion, I believe that the authors have a very valuable and interesting body of research. However, the presented results of the study need to be revised and expanded. In particular, the paper lacks an analysis of the results obtained and an explanation of the changes observed.

Author Response

The author of the submitted paper would like to thank you for reviewing this manuscript. The author greatly appreciates your comment. We have revised the manuscript based on your’ suggestions.    

Please find the attached description of all revisions. Once again, we appreciate your suggestions and hope these changes will resolve the concern. We also apologize for any revision that might be insufficient from your viewpoint.

Reviewer 2 Report

Comments and Suggestions for Authors

Author detailed an Artificial Neural Network Model’s Sensitivity Analysis: Focus on Daylight Factor, Sunlight hours, QuVue and Universal Thermal Climate Index  of residential buildings, a case study in China. The work is good however following suggestion are made to improve the quality and readability:

1) The abstract should be reorganized and should contain the main elements of this work, including the methods used and the basic conclusions obtained. So the authors should rewrite the abstract.

2) The introduction is good and literature cited are good in numbers still recommended to mentioned the work and gap observed from the literature.

3)The abstract's principal goal needs to be clarified, needs to explain why or what problem it solves

4) The introduction ends with a vague statement of the intended work's goals. Objectives should be more defined, quantifiable, and targeted to guide the study.

5) During section 2.1 Input data collection, please give the experimental work in detail.

6) Repetition and irrelevant details: Some opening lines need to be more repetitious or feature irrelevant material. Eliminating unnecessary content and streamlining information is crucial.

7) The method section needs to be more organized. Fragmented information makes it hard to follow the study method.

8) More information should be provided regarding the methodology and data collection process. It is essential to provide details about how the efficiency values were obtained, the instrumentation used, and any limitations or uncertainties associated with the data if you are okay with it.

9) How do you take into account the variables selection does it affect your results?

10) it would be necessary to explain in more detail the results obtained.

11) The discussion needs to be more comprehensive and provide a broad interpretation of the results. It is essential to relate the findings to previous studies, discuss their implications, and provide insights into the wider context. I suggest future research directions.

Author Response

(The authors gave the same response as above.)

Reviewer 3 Report

Comments and Suggestions for Authors

Dear Authors,

Article overview:

The paper Artificial Neural Network (ANN) Model’s Sensitivity Analysis: Focus on Daylight Factor (DF), Sunlight hours, QuVue (Sky View Ratio) and Universal Thermal Climate Index (UTCI) of residential buildings, a case study in China is well suited for journal Buildings.

The article presents the impact of different ANN parameter values on the quality of the obtained results.

General comments:

- The novelty of this article is low.

- The title of article is in accord with article, buy is definitely too long, in particular the repeated use of brackets with explanations needs to be changed.

- The abstract is unusually structured. The authors omitted a sentence or two of a general introduction to the issue. Then they described in too much detail what they presented in the article. Details like "MATLAB 2020a" are unnecessary, especially "2020a". The authors also presented full conclusions, this is too much. The abstract should inform, but in the form of an encouragement to read the entire article and arouse the readers' curiosity.

- Chapter 2 begins with a chart without description or explanation. This is too little. At least a short introduction should be added, indicating in which subsections the authors provided explanations for individual parts of the flowchart.

- Chapter 2.1 contains dry information about values. The authors should add explanations why such ranges of values were adopted.

- Do the authors believe that the values in Fig. 4 are useful for further simulations? What is the advantage of "randomly generated". In the reviewer's opinion, it was possible to try planned layouts based on cost considerations, typical and non-standard solutions, etc. Do the values in Fig. 5 make technical sense and do they agree with the practice of building designs?

- Page 7 – the authors adopted average values for the calculated indicators, was there another possibility in the program, is the average value the best solution?

- Fig. 7 - did the authors consider 4 tests for other configurations of the remaining parameters, or only for a fixed configuration of the remaining parameters? This is not clear to the reviewer. Studying the variability of a single ANN element may not be appropriate, the best results may be obtained with a specific number of inputs associated with a certain number of hidden neurons and also a specific number of output variables.

- Chapter 3 – whether R values this low can be considered useful at all, the authors should comment on this.

- Fig. 13 - why didn't the authors try to analyze a configuration with a much smaller number of hidden neurons? The range of values being analyzed may be completely outside the preferred range. In his work, the reviewer successfully used ANN with a small number of hidden neurons.

Detailed comments:

- The article was written enough well in English, is understandable for a reviewer, a person who does not speak English as a mother tongue. There are minor errors in the text: unnecessary spaces, missing spaces. In some places, unpopular words were used, readers may have problems understanding the authors' message. The text should be checked again.

Conclusion

Unfortunately, many results in the article reach "R" values, which should be considered "weak or no association" (below 0.4). The accepted range for the number of hidden neurons in the reviewer's opinion is not good, smaller values should also have been examined. The lack of cross-tests between models, changes in the values of various parameters, precludes obtaining results that may constitute conclusions in this work. Adopting more hidden layers was not considered. No attempt was made to reduce the number of inputs. 36 inputs means that a large amount of training data is necessary. The analysis is too basic, it is limited to assessing the variability of several parameters, and the selected values, according to the reviewer, are not in good ranges.

Comments on the Quality of English Language

- The article was written enough well in English, is understandable for a reviewer, a person who does not speak English as a mother tongue. There are minor errors in the text: unnecessary spaces, missing spaces. In some places, unpopular words were used, readers may have problems understanding the authors' message. The text should be checked again.

Author Response

(The authors gave the same response as above.)

Reviewer 4 Report

Comments and Suggestions for Authors

Some comments:

- the title is by far to bulky and unclear to read: How about Artificial Neural Network (ANN) Model's Sensitivity Analysis: Key building performance indicators for a case study in China.

- MatLab is a tool , not a method. Please utilize it that way.

- Add an abbreviation section (FFNN, RBFN, ANFIS??)-

The majority of your paper is describing the modelling and simulation, application of the mathematical/ informatory methodology onto the study case - fair enough.

What is severly missing is a critical comment toward "limitations of the study" and "future research" as well as a verification / validation approach.

Therefore, I plead for major revision.

Comments on the Quality of English Language

the language is a total mess and needs to fundamentally overworked.

E.g. abstract: " The results showed as below:" what kind of sentence is that? What about writing "The results encompass the following aspects: "

What shall "The contribution and novelty of this paper may share some guild in 95 modeling ANN to improve accuracy of building performance prediction.  " mean?

Did you use Babelfish as a translation

Author Response

(The authors gave the same response as above.)

Round 2

Reviewer 1 Report

Comments and Suggestions for Authors

Good morning,

I accept the changes made to the work. I consider them to be sufficient.

Critical remarks:

The title of the study is long and too detailed.

Line 316 - Why the page address and not the reference to the literature item?

What is missing from the paper is an assessment of the significance of the differences between the results obtained. The information that something is bigger or smaller is not sufficient. We need to show whether there is or is not a difference.

Line 555. How do you explain that for an analogous set of independent variables a model with one output is less accurate than one with several?

Author Response

Please find the attached description of all revisions. Once again, we appreciate your suggestions and hope these changes will resolve the concern. We also apologize for any revision that might be insufficient from your viewpoint.

Reviewer 2 Report

Comments and Suggestions for Authors

I accept the papaer in the updated and corrected format.

Author Response

The author of the submitted paper entitled “Artificial Neural Network (ANN) Model’s Sensitivity Analysis: Focus on Daylight Factor (DF), Sunlight hours, QuVue (Sky View Ratio) and Universal Thermal Climate Index (UTCI) of residential buildings, a case study in China” would like to thank you for reviewing this manuscript. The author greatly appreciates your comment.

Best regards,

Authors

Reviewer Comments:

Reviewer #2 I accept the papaer in the updated and corrected format.

Section

Comments from Reviewer #2

Revision

I accept the papaer in the updated and corrected format.

Thank you for the helpful comments.  

Reviewer 3 Report

Comments and Suggestions for Authors

The manuscript file after revision is not clearly legible, saving corrections made in Word in the track changes mode to PDF does not allow for the evaluation of the work, some elements are inaccessible. However, after analyzing the authors' responses to all reviewers and the revised text, my opinion is positive, although it could be better. The authors were able to respond sensibly to the reviewers' comments and introduced corrections to the text of the article.

Author Response

The author of the submitted paper entitled “Artificial Neural Network (ANN) Model’s Sensitivity Analysis: Focus on Daylight Factor (DF), Sunlight hours, QuVue (Sky View Ratio) and Universal Thermal Climate Index (UTCI) of residential buildings, a case study in China” would like to thank you for reviewing this manuscript. The author greatly appreciates your comment.

Best regards,

Authors

Reviewer Comments:

Reviewer #3

The manuscript file after revision is not clearly legible, saving corrections made in Word in the track changes mode to PDF does not allow for the evaluation of the work, some elements are inaccessible. However, after analyzing the authors' responses to all reviewers and the revised text, my opinion is positive, although it could be better. The authors were able to respond sensibly to the reviewers' comments and introduced corrections to the text of the article..

Section

Comments from Reviewer #3

Revision

Thank you for the helpful comments.

Reviewer 4 Report

Comments and Suggestions for Authors

Dear authors,

you considered the majority of my comments as such i plead for accept.

Comments on the Quality of English Language

improved, but far from perfect

Author Response

Dear Reviewers,

The author of the submitted paper entitled “Artificial Neural Network (ANN) Model’s Sensitivity Analysis: Focus on Daylight Factor (DF), Sunlight hours, QuVue (Sky View Ratio) and Universal Thermal Climate Index (UTCI) of residential buildings, a case study in China” would like to thank you for reviewing this manuscript. The author greatly appreciates your comment.

Best regards,

Authors

Reviewer Comments:

Reviewer #4

Dear authors,

you considered the majority of my comments as such i plead for accept.

Section

Comments from Reviewer #3

Revision

Thank you for the helpful comments.